# Two-dimensional bilayer ice in coexistence with three-dimensional ice without confinement

Jing Jiang ®[1], Yuanming Lai ®[1,2] ✉, Daichao Sheng[3], Guihua Tang ®[4], Mingyi Zhang[1], Dong Niu[5] & Fan Yu[1]

Icing plays an important role in various physical-chemical process. Although the formation of two-dimensional ice requires nanoscale confinement, two-dimensional bilayer ice in coexistence with three-dimensional ice without confinement remains poorly understood. Here, a critical value of a surface energy parameter is identified to characterize the liquid-solid interface interaction, above which two-dimensional and three-dimensional coexisting ice can surprisingly form on the surface. The two-dimensional ice growth mechanisms could be revealed by capturing the growth and merged of the metastable edge structures. The phase diagram about temperature and pressure vs energy parameters is predicted to distinguish liquid water, two-dimensional ice and three-dimensional ice. Furthermore, the deicing characteristics of coexisting ice demonstrate that the ice adhesion strength is linearly related to the ratio of ice-surface interaction energy to ice temperature. In addition, for gas-solid phase transition, the phase diagram about temperature and energy parameters is predicted to distinguish gas, liquid water, two-dimensional ice and three-dimensional ice. This work gives a perspective for studying the singular structure and dynamics of ice in nanoscale and provides a guide for future experimental realization of the coexisting ice.

Ice is a common state of water, which is formed by the regular arrangement of water molecules. Ice structure and nucleation play an important role in many fields such as materials science, biology, atmospheric science, cryogenic refrigeration and food engineering. Excessive ice accumulation poses great challenges to infrastructure reliability, traffic and human safety. As early as in the 1920s, Bragg, a famous British physicist and X-ray discoverer, together with other scientists, used X-ray to characterize the ice crystal structure. After one hundred years of research and exploration, 18 kinds of 3-dimensional (3D) ice crystal phases have been discovered so far. Since 1980s, predictions of various ice structures have attracted widespread attention. Over the past two decades, computer simulations have confirmed the existence of a large number of low-dimensional ice phases. The nanoscale confinement is considered as a tunable variable that can affect the overall hydrogen bond network formation of water molecules during the freezing transition. Surface ice nucleation and low-dimensional ice under extreme conditions often appear[1–4], and a large number of reported 2-dimensional (2D) ice exists on metal surfaces[5–11], insulating surfaces[12–16], graphite and graphene surfaces[14,17–22], under strong confinement conditions. Molecular dynamics (MD) simulations and experimental studies have shown that 2D crystalline and amorphous ice[20–35] appear in the supercooled state when water molecules are confined between planar hydrophobic surfaces. Water molecules can form different

[1]State Key Laboratory of Frozen Soil Engineering, Northwest Institute of Eco-Environment and Resources, CAS, Lanzhou, PR China. [2]Institute of Future Civil Technology, Chongqing Jiaotong University, Chongqing, PR China. [3]School of Civil and Environmental Engineering, University of Technology Sydney, Ultimo, NSW, Australia. [4]MOE Key Laboratory of Thermo-Fluid Science and Engineering, School of Energy and Power Engineering, Xi'an Jiaotong University, Xi'an, PR China. [5]Naval Architecture and Ocean Engineering College, Dalian Maritime University, Dalian, PR China. ✉e-mail: ymlai@lzb.ac.cn

hydrogen bond networks under nanoscale confinement. Water molecules are confined between two parallel surfaces to form single-layer and double-layer square ice[20,21], twisted double-layer hexagonal ice[22,23], double-layer hexagonal and diamond ice[24,27], double-layer interlocked pentagonal ice[28], single-layer diamond ice[29,30], three-layer ice[31], and three-layer heterogeneous fluid[32].

However, fewer studies have so far shown the possibility of 2D ice formation without nanoscale confinement. The existence of ice-like water was observed at the water-mica interface[15,36,37]. The formation mechanism of double-layer hexagonal ice (BHI) on graphene/platinum (111) substrate was revealed by the water vapor deposition experiments at 100–130 K[18]. In addition, BHI formation can be observed on Au (111), Pt (111) and Ru (0001) substrates at very low temperatures (<140 K)[9,38]. The existence of 2D ice was first confirmed by experiments combined with molecular simulation[39]. The liquid-to-BHI transition on different smooth surfaces was revealed based on molecular simulation[40,41]. In addition, the ice nucleation characteristics and de-icing mechanism of supercooled water droplets were studied by molecular simulation[42-58].

The knowledge on 2D–3D coexisting ice formation and de-icing characteristics at molecular level remains blank. The influences of the surface wettability, temperature, pressure and phase transition mode on the formation of 2D–3D coexisting ice without confinement has not been studied. Our findings fill the gap about the coexistence of 2D bilayer ice and 3D ice without nanoscale confinement. We explore the possibility of controlling the formation of low- and high-dimensional ices by tuning the interactions of a water droplet on a surface. Here, we show a groundbreaking work about the formation and growth of 2D–3D coexisting ice and de-icing characteristics in which the phase diagrams about temperature and wettability are predicted for liquid-solid and gas-solid phase transitions. Changing wettability leads to different ice growth mechanisms, which helps us to better understand water-ice behavior. This study deepens our understanding of 2D–3D coexisting ice without confinement, and is a step forward towards a predictive model for water-ice phase developments which can be used to guide and interpret future experiments. Indeed, the results presented in our work open the debate about whether it is possible to produce and control the growth of ice phases in experiments and reality.

## Results

### Correlation between the surface wettability and the contact angle

The simulation system was constructed based on LAMMPS simulation package (Supplementary Fig. 1). The smooth surface wettability is characterized by the contact angle. The correlation between the contact angle $\theta$ and the water-surface energy parameter $\varepsilon$ is clarified (Supplementary Fig. 2). As the energy parameter $\varepsilon$ increases, the contact angle decreases. For $\theta < 90°$, the solid surface is hydrophilic. For $\theta > 90°$, the solid surface is hydrophobic. For $\theta > 150°$, the solid surface is superhydrophobic. For $\theta < 15°$, the solid surface is superhydrophilic. When $\varepsilon$ exceeds the critical value, the droplet completely wets the surface with $\theta = 0°$.

### Liquid-solid phase transition: the effect of temperature on 2D ice in coexistence with 3D ice without confinement

At a constant temperature, the contact angle of nanodroplets decreases with the enhancement of nanodroplet and surface interaction, which is attributed to the larger surface energy and the stronger wettability of the solid surface (Supplementary Fig. 2). As the temperature increases, the potential energy per water molecule basically increases. The potential energy per water molecule at 250 K is significantly higher than that of per water molecule at other temperatures (Supplementary Fig. 3). In the NVT ensemble, the volume of the simulation system box is constant. As the temperature increases, the pressure of the system increases, so does the potential energy of water molecules. Figure 1a shows the change of the potential energy per water molecule with the surface energy parameters at 205 K. The potential energy per water molecule for the energy parameter $\varepsilon \leq 0.43$ kcal·mol$^{-1}$ is significantly higher than that of the energy parameter $\varepsilon > 0.43$ kcal·mol$^{-1}$. As energy parameter $\varepsilon$ increases, the potential energy suddenly drops by -0.43 kcal·mol$^{-1}$. The potential energy declines sharply between $\varepsilon = 0.40$ kcal·mol$^{-1}$ and $\varepsilon = 0.50$ kcal·mol$^{-1}$. Figure 1b displays the mean-square displacement of water molecules on the solid surface at 205 K. The mean-square displacement of $\varepsilon = 0.50$ kcal·mol$^{-1}$ corresponds to the vertical axis on the right-hand side, which is one order of magnitude higher than that of $\varepsilon = 0.43$ kcal·mol$^{-1}$ and 0.30 kcal·mol$^{-1}$ corresponds to the vertical axis on the left-hand side. The mean square

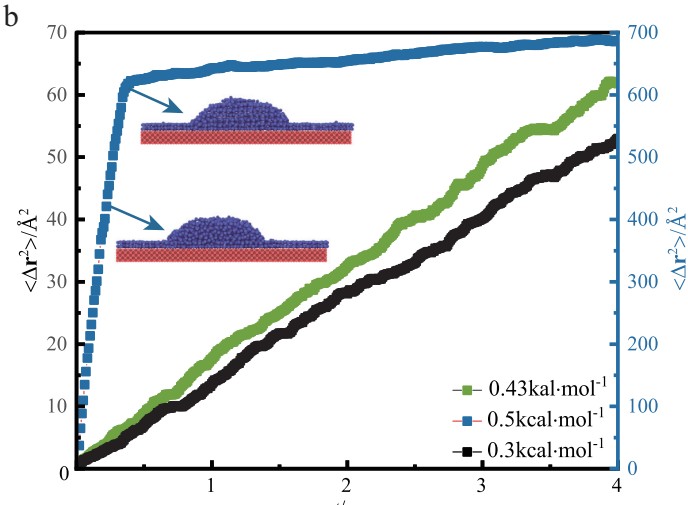

**Fig. 1 | Determination of energy parameter for phase transition. a** Potential energy per water molecules versus energy parameter $\varepsilon$ on the Pt surface at $T = 205$ K after equilibrium. Data are presented as mean values +/− SD (standard deviation), $n = 20$. **b** Mean-square displacement <$\Delta r^2$> of water molecules versus time $t$ on the solid surface at $T = 205$ K. r is the positions of atom. The mean-square displacement of $\varepsilon = 0.50$ kcal·mol$^{-1}$ corresponds to the right-hand vertical axis. The mean-square displacement of $\varepsilon = 0.43$ kcal·mol$^{-1}$ and 0.30 kcal·mol$^{-1}$ corresponds to the left-hand vertical axis.

displacement of $\varepsilon = 0.50$ kcal·mol⁻¹ increases sharply at first, showing some water molecules of the droplet diffuse rapidly along the wall to form 2D ice. In this process, the liquid water exists at the same time as the 2D ice solid. Then the inflection point appears and tends to increase slowly, indicating that the liquid water molecules nucleate and grow into 3D ice. At this time, both 2D ice solid phase and 3D ice solid phase are present. The whole freezing process has two phase transitions. The first phase transition produces 2D ice, and the second phase transition produces 3D ice. The appearance of the inflection point means that the liquid water molecules nucleate and crystallize in an instant to form 3D ice, and only the solid state exists. For the mean square displacement of $\varepsilon = 0.43$ kcal·mol⁻¹, there is only liquid phase after equilibrium. The critical value of energy parameter $\varepsilon$ for phase transition is estimated to be 0.43 kcal·mol⁻¹.

The dynamic processes of nanodroplets on surfaces with different energy parameters show characteristics of the nanodroplets supercooled to nucleation and crystallization (Supplementary Fig. 4 and Fig. 5). The nucleation process is a phase-transition activation process. Overcoming the free energy barrier to form critical nucleus, the nucleus grows spontaneously (Supplementary Fig. 6). The phase transition and crystallization process of nanodroplets on surfaces with different energy parameters can be summarized as: (1) Droplets supercooled; (2) Critical nucleus formation; (3) Rapid phase transition; (4) Ice crystals growth. Near the gas-liquid, liquid-solid and gas-liquid-solid regions, the appearance of critical nucleus is spatially selective. The simulation results show that the surface nucleation process with different wettabilities after quenching displays different characteristics: For $\varepsilon = 0.10$ kcal·mol⁻¹, the formation of critical nucleus preferentially occurs inside the droplet rather than on the solid surface, which is attributed to the superhydrophobic state of the surface ($\theta > 150°$). The extremely low free energy of the solid surface causes the migration of nanodroplet. For $\varepsilon = 0.20$–$0.43$ kcal·mol⁻¹, nanodroplet nucleation occurs preferentially on solid surface due to the surface being between hydrophobic and hydrophilic state ($70° < \theta < 150°$). With the increase of wettability, the interaction between nanodroplet and the surface enhances, so the contact angle of nanodroplet decreases and the contact radius increases. Nanodroplet on surface with higher wettability is more likely to nucleate and crystallize due to the very high adsorption energy of the surface. Nanodroplet crystallization is not a single cubic or hexagonal structure, but a mixture of cubic and hexagonal structure stacking on each other. The stacking direction is controlled by the growth direction of ice nucleus. In the process of water freezing, a relatively long-lived hydrogen-bonded network is generated between water molecules. For the liquid-solid phase transition, from the perspective of dynamics, the decrease in temperature leads to slower molecular motion and lower kinetic energy. When the kinetic energy of the molecule is small, the potential energy between the molecules is sufficient to confine the molecules, so that the molecules are usually arranged in a regular crystal structure. The nucleation and crystallization of nanodroplets mainly occur on the solid surface ($\varepsilon = 0.20$–$1.0$ kcal·mol⁻¹), and the latent heat released by local crystallization can provide enough energy for growth and crystallization[52]. The variation of the potential energy with time displays the nucleation and crystallization process of nanodroplets after quenching and cooling (Supplementary Fig. 7). The nucleation event is detected by recording the sudden decrease of the potential energy $E_{pot}$ of water molecules during the formation of critical ice nucleus. For $\varepsilon = 0.10$–$0.43$ kcal·mol⁻¹, the potential energy change of the nucleation and crystallization process of nanodroplets on the solid surface is divided into three continuous characteristic stages: In the first stage, the potential energy of water molecules changes slightly. In the second stage, the potential energy $E_{pot}$ of water molecules decreases sharply, and the critical nucleus is formed. In the third stage, the potential energy of water molecules is in an equilibrium, and the remaining water molecules slowly crystallize.

For 0.43 kcal·mol⁻¹ $< \varepsilon \leq 1.0$ kcal·mol⁻¹, the nanodroplets first form 2D ice and then form 3D ice during the supercooled process. Taking $\varepsilon = 0.50$ kcal·mol⁻¹ as an example, the dynamic process of the formation of 2D-3D coexisting ice is shown in Fig. 2a. After quenching and cooling, some water molecules on the surface droplets diffuse rapidly to form 2D ice. The 2D ice is formed by two layers of interlocking 5-, 6-, 7- membered rings water molecules. The double-layers are connected by hydrogen bonds. Each water molecule forms three hydrogen bonds with water molecules in the same layer, and forms a hydrogen bond with water molecules in the upper and lower layers. Therefore, all hydrogen bonds are saturated and their structures are very stable[39]. 2D ice formation mechanism without nanoscale confinement is that appropriate water-surface interactions can compensate for the entropy loss in the freezing transition process[41]. The supercooled unfrozen droplets are promoted by 2D ice. After overcoming the free energy barrier and forming a critical nucleus, the crystal nucleus grows rapidly and crystallizes to form 2D and 3D coexisting ice. The 3D ice is formed by the mixed accumulation of hexagonal ice and cubic ice. The 2D ice underlying growth mechanisms could be revealed by capturing the growth and merged of the metastable edge structures (Supplementary Fig. 8). Firstly, the edges of the 2D ice on both sides are formed by the non-rotating stacking of double-layer 4-, 5-, 6-, 7-membered ring ice. Then, with the diffusion of water molecules, the 2D ice growth edge is composed of double-layer 5-, 6-, 7-membered ring water molecules. The 2D ice boundary has metastable characteristics. At the moment of merging at the edge of the 2D ice double layer, the 6-membered ring in the white circle in the picture contributes a single water molecule, which is connected with the left 5-membered ring, and then forms two metastable 4-membered rings to connect the water molecules on both sides. After that, the two 4-membered ring water molecules become relatively stable 5-membered ring water molecules, and the water molecules on both sides merge and grow. During the growth process, the zigzag growth mode and the armchair growth mode of the 2D ice appears. When the calculation time reaches 200 ns, the 2D-3D coexisting ice structures are relatively stable. The 2D ice is still composed of double-layer 5-, 6-, 7-membered ring water molecules, and the proportion of 5-, 7-membered ring water molecules is basically unchanged, indicating that the 2D-3D coexisting ice is not sensitive to the calculation time when it is sufficiently long (Supplementary Fig. 9). The intensity of the first peak is slightly larger than that of the second peak as shown in Fig. 2b, revealing that the growth of 2D ice starts from the bottom layer (near the solid wall), which is consistent with the results in ref. 39. Figure 2c shows the growth of the number of ice molecules. After quenching and cooling, 2D ice grows rapidly and its number reaches a stable value. After that, the unfrozen droplets overcome the free energy barrier to form 3D ice quickly, and the number of 2D ice and 3D ice molecules reaches the maximum.

Based on the molecular simulation calculation, the critical value of the surface energy parameter is estimated to be 0.43 kcal·mol⁻¹. The evaluation system of liquid water, 2D ice and 3D ice on the temperature and energy parameter $\varepsilon$ on the Pt surface is predicted. As shown in Fig. 2d, the influence of multiple temperature conditions and multiple energy parameters on the liquid-solid phase transition of water molecules is clarified. For 210 K $< T \leq 250$ K and $\varepsilon \leq 0.43$ kcal·mol⁻¹, the supercooled nanodroplet is in liquid phase without phase transition. For 210 K $< T \leq 250$ K and $\varepsilon$ å 0.43 kcal·mol⁻¹, the supercooled nanodroplet is in the coexistence of 2D ice and liquid phase. For 130 K $\leq T \leq 210$ K and $\varepsilon \leq 0.43$ kcal·mol⁻¹, the supercooled nanodroplet is in solid phase with phase transition, and the 3D ice is disorderly accumulated by hexagonal ice and cubic ice. For 130 K $\leq T \leq 210$ K and $\varepsilon$ å 0.43 kcal·mol⁻¹, the supercooled nanodroplets undergoes phase transition, and the ice is in the coexistence of 2D ice and 3D hexagonal ice and cubic ice. The mechanism for the 2D-3D coexisting ice without nanoscale confinement is attributed to overcoming the free energy barrier and the satiation of the Bernal-Fowler ice rules and the

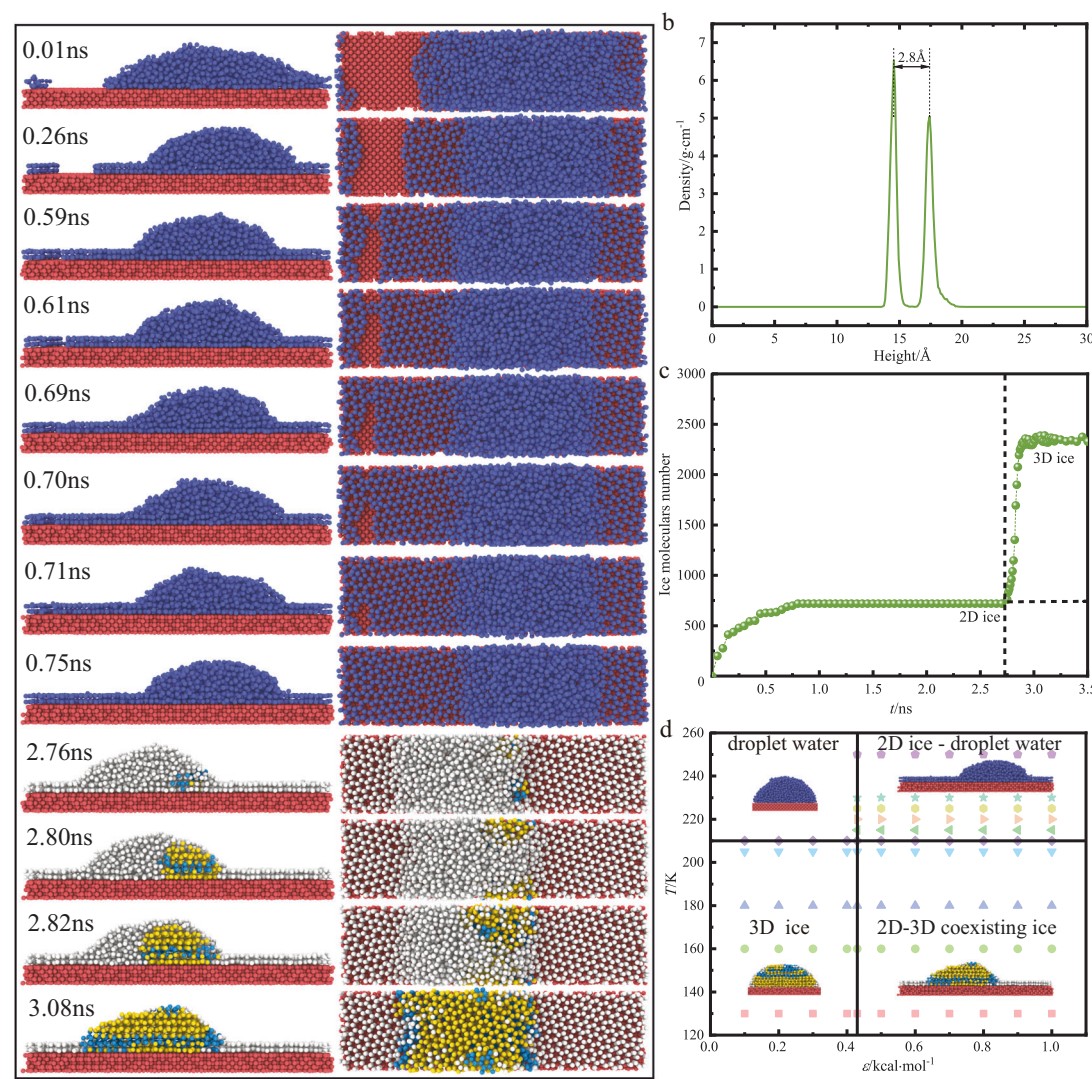

**Fig. 2 | Growth characteristics of 2D ice in coexistence with 3D ice. a** A sequence of snapshots of the coexisting ice growth process (Front view and top view) for $\varepsilon = 0.5$ kcal·mol$^{-1}$. Dark yellow, light blue, light gray (3.08 ns) and light red balls represent hexagonal ice, cubic ice, 2D ice and solid substrate, respectively. **b** Density profile of the 2D ice. Density represents the size of 1D density. Height is the size of $z$ axis. **c** The number of ice molecules versus time. **d** Phase diagram of liquid water, 2D ice and 3D ice with respect to temperature $T$ and energy parameter $\varepsilon$. Phase diagram consists of liquid water, 2D ice in coexistence with liquid water, 3D ice and 2D-3D coexisting ice.

appropriate water-surface interaction, which can compensate for the entropy loss caused by the freezing transition process.

## Liquid-solid phase transition: the effect of pressure on 2D ice in coexistence with 3D ice without confinement

In order to investigate the influence of pressure on coexisting ice, five pressure conditions were chosen for molecular dynamic simulation at 205 K. In the NVT ensemble, pressure regulation is achieved by adding different numbers of nitrogen molecules (Supplementary Fig. 10). Due to the limited size of the system box, 20, 50, 100, 200 and 300 nitrogen molecules were added to obtain approximate pressures of 1.0, 2.5, 5.5, 11.0 and 17.5 atmospheres, respectively. Figure 3b shows that the contact angle of nanodroplets decreases with increasing gas pressure, and the droplet curvature decreases first and then increases with increasing gas pressure for energy parameter $\varepsilon = 0.4$ kcal·mol$^{-1}$. Taking pressure of 11.0 atm as an example, for 0.1 kcal·mol$^{-1} \leq \varepsilon < 0.43$ kcal·mol$^{-1}$, the liquid-solid phase transition only generates 3D ice with disordered stacking of hexagonal ice and cubic ice (Supplementary Fig. 11). For 0.43 kcal·mol$^{-1} \leq \varepsilon \leq 1.0$ kcal·mol$^{-1}$, the liquid-solid phase transition

produces coexisting ice, which is composed of 2D ice and 3D ice stacked disorderly with hexagonal ice and cubic ice (Supplementary Fig. 12). With increasing wetting characteristics, the contact angle of droplets decreases, and droplets gradually spread on the wall. After quenching, 2D ice begins to form instantaneously, and 3D ice is formed after overcoming the nucleation energy barrier. The variation of potential energy with time also reflects the occurrence of nucleation and icing process (Supplementary Fig. 13). When the surface energy parameters is 0.5 kcal·mol$^{-1}$, 2D ice grows and merges slowly for pressure 11.0 atm (and 17.5 atm) as shown in Fig. 3a, which is attributed to the combined effect of surface wettability and gas pressure. After calculation of 200 ns, the 2D ice is still composed of double-layer 5-, 6-, 7-membered ring water molecules.

The phase diagram about energy parameters and pressure is predicted to distinguish 3D ice and 2D-3D coexisting ice by a large number of molecular dynamic simulations. The critical value of the surface energy parameter is estimated to be 0.43 kcal·mol$^{-1}$, which is the same as that of the nitrogen-free system, indicating that the formation of 2D-3D coexisting ice on the platinum surface is not very sensitive to the applied gas pressure conditions. The effects of

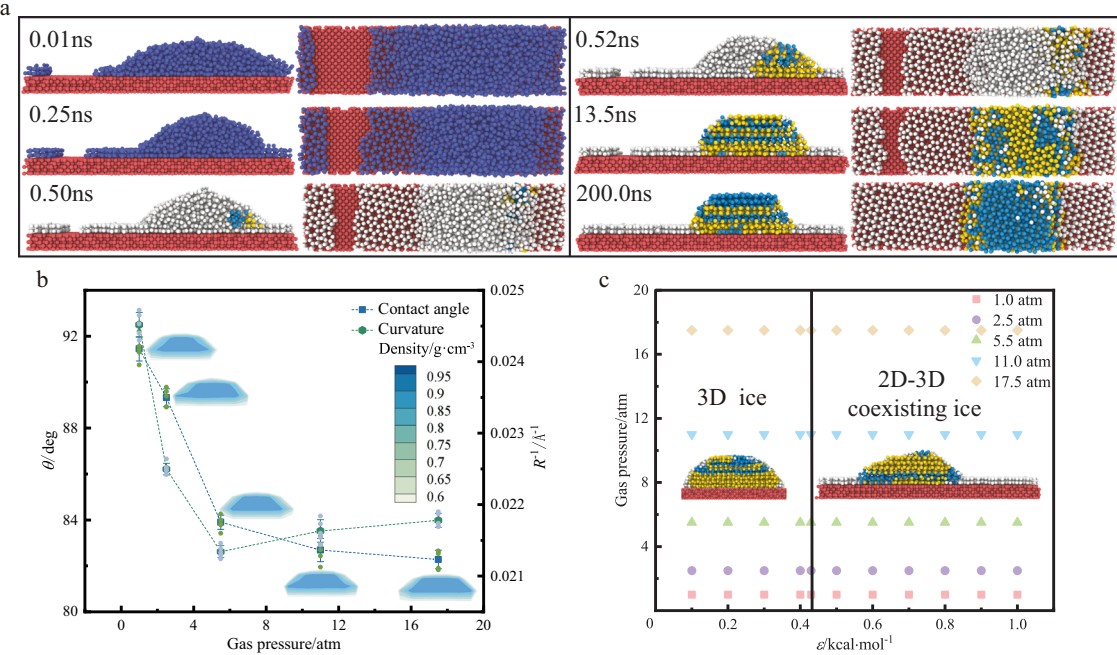

**Fig. 3 | Effect of gas pressure on growth characteristics of 2D-3D ice. a** A sequence of snapshots of the coexisting ice growth process (Front view and top view) for gas pressure $P = 11$ atm and energy parameter $\varepsilon = 0.5$ kcal·mol⁻¹. Dark yellow, light blue, light gray (200.0 ns) and light red balls represent hexagonal ice, cubic ice, 2D ice and solid substrate, respectively. **b** Variation of contact angle $\theta$ and curvature $R^{-1}$ versus pressure for $\varepsilon = 0.4$ kcal·mol⁻¹. A density colorbar shows the meaning of the contact angle heatmaps. Data are presented as mean values +/− SD (standard deviation), $n = 5$. **c** Phase diagram of 3D ice and coexisting ice with respect to gas pressure and energy parameter $\varepsilon$. Phase diagram consists of 3D ice and 2D-3D coexisting ice.

multiple pressure conditions and multiple surface wetting characteristics on the liquid-solid phase transition of water molecules are revealed as shown in Fig. 3c. For 1.0 atm ≤ $P$ ≤ 17.5 atm and $\varepsilon < 0.43$ kcal·mol⁻¹, the supercooled droplet undergoes liquid-solid phase transition to solid phase, and the 3D ice is composed of disordered hexagonal ice and cubic ice. For 1.0 atm ≤ $P$ ≤ 17.5 atm and 0.43 kcal·mol⁻¹ ≤ $\varepsilon$ ≤ 1.0 kcal·mol⁻¹, liquid-solid phase transition occurs, and the coexisting ice consists of 2D ice and 3D hexagonal ice and cubic ice. For 0.43 kcal·mol⁻¹ ≤ $\varepsilon$ < 0.8 kcal·mol⁻¹, the 2D ice in coexisting ice is composed of double-layer 5-, 6-, 7-membered ring water molecules. For 0.8 kcal·mol⁻¹ ≤ $\varepsilon$ ≤ 1.0 kcal·mol⁻¹, the 2D ice in coexisting ice is finally composed of a layer of 5-, 6-, 7-membered ring water molecules and a layer of 4-, 6-membered ring water molecules near solid surface (Supplementary Fig. 14). The appearance of the 4-membered square ring water molecules is attributed to the surface being too hydrophilic and gas pressure.

## De-icing mechanism of 2D ice in coexistence with 3D ice

The detachment processes of nanoscale coexisting ice from smooth Pt surface are used to investigate the de-icing characteristics (Supplementary Fig. 15). Ice adhesion strength is the ice adhesion force divided by the ice-solid substrate contact area ($\sigma = F / A$). The tension is applied to the ice by applying acceleration to all water molecules in the +z direction (perpendicular to the solid substrate), increasing the vertical acceleration at a constant rate, so that the tension is stably loaded throughout the ice. With the superposition of force fluctuations, the attractive force between the coexisting ice and the solid substrate increases linearly (Supplementary Fig. 16a). When the coexisting ice is completely detached from the solid substrate instantaneously, the attractive force decreases suddenly. The coordinates of the intersection point of fluctuating attractive force fitting curve (the black line) and the expected attractive force (the red line) are the detachment force and detachment time (Supplementary Fig. 16b). When the

tension is greater than the maximum adhesion between the ice and the solid substrate, the ice will be completely detached from the solid substrate, and the ice detachment is related to the sudden decrease of the adhesion force. MD simulation is able to capture the ice detachment process at the atomic level[55], and provide nanoscale mechanism of ice detachment dynamics. Figure 4a reveals that the ice adhesion force of 2D ice in coexistence with 3D ice is significantly larger than that of 3D ice. Figure 4b shows that the adhesion strength of 2D-3D coexisting ice is smaller than that of 3D ice, which is attributed to the combined effect of ice adhesion force and ice contact area. Ice adhesion strength depends on the ice adhesion force and the ice contact area. The contact area between 2D-3D coexisting ice and solid substrate is 2.35 times that between 3D ice and solid substrate ($A_{\text{2D-3D}} / A_{\text{3D}} = 2.35$). The 2D-3D coexisting ice has a larger contact area with a solid substrate, which could provide a higher adhesion force. However, the adhesion force of 2D-3D coexisting ice is no more than 2 times that of 3D ice ($F_{\text{2D-3D}} / F_{\text{3D}} < 2$). Therefore, ice adhesion strength for 2D-3D coexisting ice is lower than that of 3D ice. Figure 4c, d shows the dynamic process of coexisting ice and 3D ice detachment from the solid substrate, respectively. The coexisting ice is first detached from the surface at the left junction of the 2D ice and the 3D ice, then the 2D ice and the 3D ice on the left side are detached from the surface, and finally the 2D ice on the right side is detached from the surface. The 3D ice is first detached from the surface at the right edge, becoming increasingly tilted relative to the substrate and eventually detached from the substrate. This angular ice detachment mechanism is consistent with the experimental observation results[54], which is attributed to the sharp edge or corner of the ice having the maximum tensile stress due to its singularity. Therefore, the thermal fluctuation of ice molecules is most likely to cause ice to break away from the edge corner. The calculated ice adhesion strength is two orders of magnitude larger than the experimental ice adhesion strength (≤ 1 MPa), which is mainly attributed to the fact that the typical loading rate used

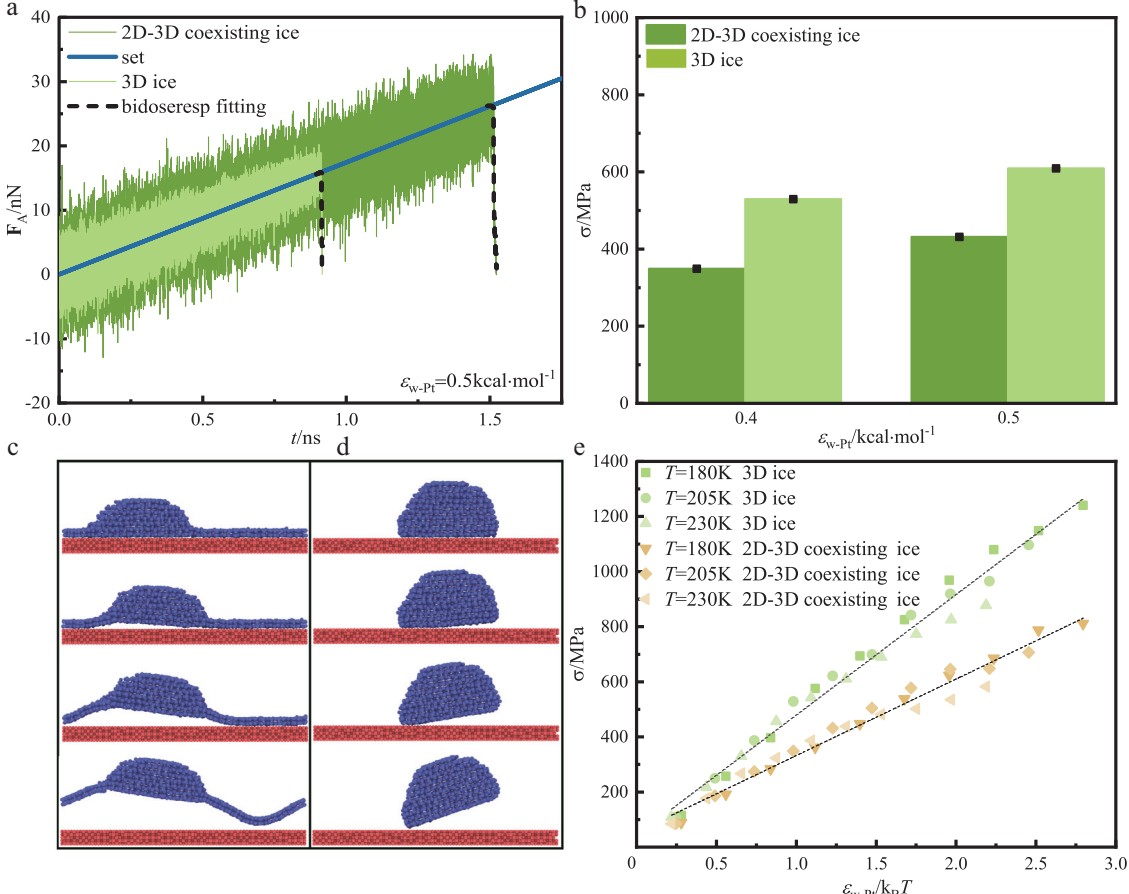

**Fig. 4 | Comparison of ice adhesion strength between the 2D-3D coexisting ice and 3D ice. a** Correlation between fluctuating adhesion force and time for 2D-3D coexisting ice and 3D ice detached from smooth Pt surface. $\mathbf{F}_A$ is the fluctuating adhesion force between ice and solid substrate. Set is the pulling force acting on ice as a function of time. It represents that the initial value of acceleration is 0 and increases linearly with $5.734 \times 10^{-9}$ nm·fs$^{-2}$. Bidoseresp fitting is the least-square fitting of the datas for the part below the "set" line using the bidirectional-dose- response function. $\varepsilon_{w\text{-Pt}}$ means the energy parameter between the 2D ice or 2D-3D ice and the walls. **b** Ice adhesion strength σ vs energy parameter $\varepsilon$. Data are presented as mean values +/- SD (standard deviation), $n = 3$. **c** A sequence of snapshots of the 2D-3D coexisting ice detachment process. **d** A sequence of snapshots of the 3D ice detachment process. **e** Variation of ice adhesion strength versus the ratio $\varepsilon_{w\text{-Pt}}/(k_B T)$ for the 2D-3D coexisting ice and 3D ice systems. The dash lines are obtained by linear fitting of the data.

in MD simulation is about 6–7 orders of magnitude higher than the experimental study[55], the nano-size of ice and the separation stress of ice increase logarithmically with the loading rate[56]. If the same loading rate is used in MD simulation and experiment, the de-icing stress of the two methods is not much different.

The adhesion strength of coexisting ice is significantly lower than that of 3D ice for different temperatures and the same energy parameters (Supplementary Fig. 17). At the same temperature, the adhesion strength of 3D ice and coexisting ice is linearly related to the energy parameter $\varepsilon_{w\text{-Pt}}$ between nanodroplets and solid substrate, indicating that the surface wettability energy parameter $\varepsilon_{w\text{-Pt}}$ is directly related to the ice adhesion strength. Based on the concept of depletion layer, the linear correlation between $\cos\theta$ and $\varepsilon$ of water droplets on the solid substrate is derived[57]. By assuming the homogeneous solid density $\rho_S$ and liquid density $\rho_L$, and ignoring the electrostatic or interfacial entropy, the work $H_{12}$ of each surface area required to separate the liquid from the solid is calculated theoretically.

$$H_{12} = \gamma_{SV} + \gamma_{LV} - \gamma_{SL} = -\pi\rho_L\rho_S \int_{z^*}^{R_0} dz z(z - z^*)^2 u(z) \quad (1)$$

$$1 + \cos\theta = (\gamma_{SV} + \gamma_{LV} - \gamma_{SL})/\gamma_{LV} = H_{12}/\gamma_{LV} \sim \varepsilon_{W-Pt} \quad (2)$$

$$\sigma \sim 1 + \cos\theta \quad (3)$$

$$1 + \cos\theta^* = \phi_s(1 + \cos\theta) \quad (4)$$

where $\gamma_{SV}$, $\gamma_{LV}$ and $\gamma_{SL}$ are the surface tensions between solid-vapor (SV), liquid-vapor (LV) and solid-liquid (SL), respectively. $u(z)$ represents the solid-liquid interaction energy. The LJ interaction simulation analysis is used. Since the LJ potential is linearly related to $\varepsilon$, $H_{12}$ is also a linear function of $\varepsilon$. Equation (2) is obtained by combining Young's equation. The ice adhesion strength is linearly related to the cosine of the receding angle of water droplets on the solid surface[58], that is, $\sigma \sim 1 + \cos\theta_{rec}$. The atomic plane does not exhibit contact angle hysteresis, indicating that the receding angle of the water droplet is the same as the equilibrium contact angle. Therefore, the ice adhesion strength and the linear variation of the contact angle cosine of water droplets on the solid surface is given by Eq. (3). From Eqs. (2) and (3), it is obtained that the ice adhesion strength is a linear function of the water-solid substrate interaction force $\sigma \sim \varepsilon$. The slope of linear function of coexisting ice adhesion strength and energy parameter $\varepsilon$ on solid surface is smaller than that of linear function of 3D ice adhesion strength and energy parameter $\varepsilon$ (Supplementary Fig. 17). As the interaction energy between ice and solid substrate decreases, the ice adhesion strength of solid surface decreases.

The ice temperature and the ice-solid substrate interaction force are important parameters affecting the ice adhesion strength. The increase of temperature enhances the fluidity of water molecules in ice, which reduces the ice adhesion strength. Temperature and ice-solid substrate interaction have opposite effect on ice adhesion strength. For a given ice-solid substrate interaction, higher temperature corresponds to lower ice adhesion strength. For different temperatures, the ice adhesion strength is still linearly related to the surface energy parameter. Considering the opposite effects of temperature and water-substrate interaction on ice adhesion strength, Fig. 4e shows the linear relationship between ice adhesion strength and dimensionless number $\varepsilon/k_B T$. The linear correlation slope of $\varepsilon/k_B T$ of coexisting ice on solid surface is smaller than that of 3D ice on solid surface, which is consistent with the results shown in Supplementary Fig. 16. With the decrease of ice-solid substrate interaction, the ice adhesion strength on solid surface decreases. At different temperatures, the ice adhesion strength is linearly related to the ice-solid surface interaction. For the coexisting ice and 3D ice on the solid surface, the MD data of ice adhesion strength are fitted to obtain two straight lines, namely, the ice-solid substrate interaction force and ice temperature ratio function.

In NPxyT ensemble, after 200 ns of calculation, the 2D ice composed of metastable double-layer 5-, 6-, 7-membered ring water molecules is transformed into steady-state 6-membered ring 2D ice. 3D ice structure in coexisting ice is not sensitive to the calculation time (Supplementary Fig. 18). At 30 ns, the 2D ice is composed of double-layer 5-, 6-, 7-membered ring water molecules. At 200 ns, the 2D ice is composed of double-layer 6-membered ring water molecules. With the increase of energy parameter $\varepsilon_{w-Pt}$, the ice adhesion force at 200 ns is gradually larger than that at 30 ns, which is attributed to the combined effect of surface wetting characteristics, 2D ice structure and ice freezing time (Supplementary Fig. 19).

## Gas-solid phase transition: 2D ice in coexistence with 3D ice without confinement

A gas-liquid-solid three-phase system is established to simulate vapor deposition (Supplementary Fig. 20). The dynamic processes of vapor deposition display the characteristics of nucleation and crystallization (Supplementary Fig. 21 and Fig. 22). For different surface wettabilities, dropwise condensation mode or film condensation mode can appear in supercooled process. For $\varepsilon = 0.10 - 0.20 \, kcal \cdot mol^{-1}$, water vapor is difficult to deposit on the solid surface because the surface is too hydrophobic. For $\varepsilon = 0.30 - 0.40 \, kcal \cdot mol^{-1}$, with the progress of vapor deposition, the gas-liquid phase variation occurs, and independent nanodroplets appear on the solid surface. The water vapor condenses in dropwise condensation mode, and then the liquid-solid phase transition occurs. The nucleation of the nanodroplet preferentially occurs on the solid surface. Nanodroplet crystallization is a stacking disordered arrangement of cubic and hexagonal structure. For $\varepsilon = 0.43-0.50 \, kcal \cdot mol^{-1}$, the solid surface is hydrophilic. With the progress of vapor deposition, the gas-solid phase transition occurs, and the gas phase molecules diffuse to form 2D ice. The 2D ice is composed of double-layer 5-, 6-, 7-membered ring ice without rotation and stacking. Then the gas-liquid phase transition occurs, and a large number of gas molecules condense in the dropwise condensation mode. With the progress of vapor deposition, the liquid-solid phase transition is activated. The free energy barrier is overcome to form a critical nucleus. The critical nucleus is close to the 2D ice, and the crystal nucleus grows rapidly and crystallizes to form 3D stacking ice with hexagonal structure and cubic structure. 2D ice promotes the nucleation and growth of 3D ice, thus forming 2D and 3D coexisting ice. The formation mechanism of 2D ice without confinement is attributed to the fact that appropriate water-surface interactions can compensate for the entropy loss[41]. For $\varepsilon = 0.60 \, kcal \cdot mol^{-1}$, the solid surface is hydrophilic. Firstly, the gas-solid phase transition occurs to form 2D ice. Secondly, the gas-liquid phase transition occurs and condenses in dropwise condensation mode. Then, the liquid-solid phase transition occurs, the crystal nucleus grows rapidly and crystallizes, and the gas-solid phase transition occurs at the same time. The gas molecule directly becomes a component of the 3D ice, and finally only 3D ice is formed. For $\varepsilon = 0.70 - 1.0 \, kcal \cdot mol^{-1}$, firstly, the gas-solid phase transition occurs to form 2D ice, and then the gas-liquid phase transition occurs and condenses in dropwise condensation mode. Secondly, 2D ice is covered by a complete liquid film, showing the film condensation mode. Finally, the liquid-solid phase transition occurs, forming 3D hexagonal structure and cubic structure disordered accumulation ice. For $\varepsilon = 0.10-0.20 \, kcal \cdot mol^{-1}$, the potential energy of gas molecules keeps in equilibrium without phase transition. For $\varepsilon = 0.30 - 1.0 \, kcal \cdot mol^{-1}$, the potential energy of gas molecules decreases twice (Supplementary Fig. 23). The first drop is gas-liquid phase transition or the gas-solid phase transition coexists with gas-liquid phase transition, and the second drop is liquid-solid phase transition, which verifies the different surface condensation crystallization phase transition characteristics mentioned above.

For $0.43 \, kcal \cdot mol^{-1} \leq \varepsilon \leq 1.0 \, kcal \cdot mol^{-1}$, the vapor deposition first forms 2D ice during the supercooled process, and then forms 3D ice. For $\varepsilon = 0.50 \, kcal \cdot mol^{-1}$, the formation process of 2D and 3D coexisting ice is shown in Fig. 5a. As the vapor deposition progresses, the gas-solid phase transition occurs. The gas molecules first form a single 2D double-layer 5-membered ring ice on the solid surface, then form a dispersed 2-dimensional ice on the solid surface composed of the non-rotating stacking of double-layer 5-, 6-, 7-membered ring ice. The dispersed 2D ice merge and grow. The gas molecules are condensed in dropwise condensation mode. The critical nucleus forms and grows rapidly to form 3D ice. Finally, 2D and 3D coexisting ice is formed. Figure 5b shows the relationship between temperature and potential energy. The first potential energy drop corresponds to the coexistence of 2D ice and nanodroplet, including gas-solid phase transition and gas-liquid phase transition. The second potential energy drop corresponds to the formation of 3D ice critical nucleus, including only liquid-solid phase transition. The second potential energy drop corresponds to the temperature jump, indicating that the heat released by local nucleation crystallization can increase the temperature.

Through a large number of the molecular dynamics simulations, the critical value of surface energy parameter is estimated to be $0.43 \, kcal \cdot mol^{-1}$. The evaluation system of temperature and energy parameter $\varepsilon$ of gaseous water, liquid water, 2D ice and 3D ice is predicted as shown in Fig. 5c. For $200 \, K \leq T \leq 250 \, K$ and $\varepsilon \leq 0.20 \, kcal \cdot mol^{-1}$, there is no phase transition. For $200 \, K \leq T \leq 210 \, K$ and $0.20 < \varepsilon < 0.43 \, kcal \cdot mol^{-1}$, the gas-liquid-solid phase transition occurs. The system only has dropwise 3D ice. For $200 \, K \leq T \leq 210 \, K$ and $0.43 \leq \varepsilon < 0.60 \, kcal \cdot mol^{-1}$, the system has 2D and 3D coexisting ice. For $200 \, K \leq T \leq 210 \, K$ and $0.60 \leq \varepsilon \leq 1.0 \, kcal \cdot mol^{-1}$, the gas-liquid-solid phase transition occurs. The system only has film-like 3D ice. For $210 \, K < T \leq 250 \, K$ and $0.20 < \varepsilon < 0.60 \, kcal \cdot mol^{-1}$, the system exists 2D ice and dropwise droplet. For $210 \, K < T \leq 250 \, K$ and $0.60 \leq \varepsilon \leq 1.0 \, kcal \cdot mol^{-1}$, 2D ice and film-like water coexist in the system.

## Discussion

In summary, we have investigated the effect of surface wettability and temperature on the formation mechanism and de-icing characteristics for 2D-3D coexisting ice. We find that 2D-3D coexisting ice can form from liquid water without nanoscale confinement. The surface wettability plays an important role in the liquid phase to coexisting ice phase transition. A critical energy parameter value has been identified, above which the transition from liquid to 2D-3D coexisting ice phase can be obtained. The 2D ice growth mechanisms could be revealed by capturing the growth and merged of the metastable edge structures. The phase diagram about temperature

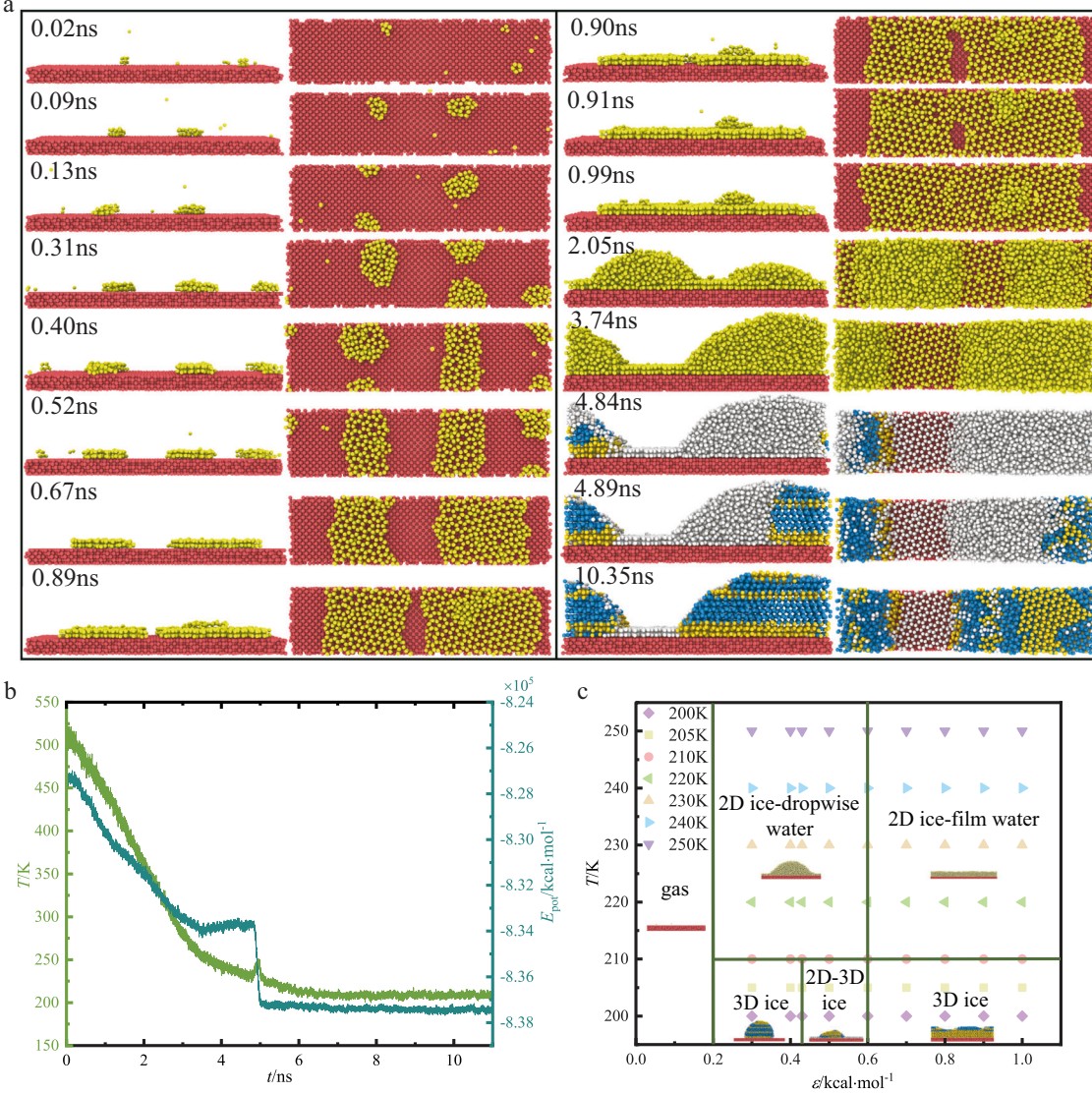

**Fig. 5 | Growth characteristics of 2D ice in coexistence with 3D ice. a** A sequence of snapshots of the coexisting ice growth process (Front view and top view) for $\varepsilon = 0.5$ kcal·mol$^{-1}$. $\varepsilon$ is the energy parameter between the water vapor and the wall. Dark yellow, light blue, light gray (10.35 ns) and light red balls represent hexagonal ice, cubic ice, 2D ice and solid substrate, respectively. **b** Variation of the temperature $T$ and potential energy $E_{pot}$ versus time $t$. **c** Phase diagram criterion of gaseous water, liquid water, 2D ice and 3D ice regarding temperature $T$ and energy parameter $\varepsilon$. Phase diagram consists of gas, 2D ice in coexistence with dropwise liquid water, 2D ice in coexistence with film liquid water, 3D dropwise ice, 3D film ice and 2D-3D coexisting ice.

and pressure vs energy parameters is predicted to distinguish liquid water, 2D ice and 3D ice. Furthermore, we have demonstrated that the ice adhesion strength is linear with the ratio of the ice-surface interaction energy to the ice temperature. In addition, for gas-to-solid phase transition, the phase diagram about temperature and energy parameters is predicted to distinguish gas, liquid water, 2D ice and 3D ice. The findings provide a molecule-level understanding of the formation mechanism and de-icing characteristics of 2D-3D coexisting ice without confinement, which is of great significance for understanding the ice structure and dynamics in nanoscale. Our work will hopefully motivate future experimental studies of the formation of low- and high-dimensional ice on a variety of realistic surfaces. The change of the $\varepsilon$ parameter could potentially be achieved through modifications of lattice planes or surface characteristics. The adjustment of surface characteristics is achieved by adjusting the wettability or constructing rough structures. 2D ice on the surface can promote or inhibit the formation of 3D ice, which has potential application value for the design and development of anti-icing materials.

## Methods

### Liquid-solid phase transition: MD simulation of nanodroplet nucleation

The nanodroplet-solid surface simulation system was constructed with a platinum surface as the substrate (Supplementary Fig. 1). The research system contains 2430 water molecules and 4000 platinum atoms. The initial nanodroplets are located on the upper part of the rough surface of the metal platinum, and the platinum atoms with a lattice constant of 3.92 Å are arranged in a face-centered cubic (FCC). The $x$ and $y$ directions are periodic boundary conditions, the $z$ direction is non-periodic boundary conditions, and the reflection boundary conditions are set at the top. The size of simulated box is $15.69 \times 3.92 \times 11.77$ nm. To improve the calculation efficiency and reduce the calculation time and cost, the coarse-grained water molecule model (mW)[42] is used to describe the interaction force between water molecules in nanodroplets, and the interaction force between solid wall and water molecules in nanodroplets is described by 12-6 LJ interaction potential. The energy parameter $\varepsilon$ ranges from 0.1 to 1.0 kcal·mol$^{-1}$ by adjusting the surface wettability energy

parameters between solid surface and water molecules, covering the solid surface from superhydrophobicity to superhydrophilicity. The interaction between platinum atoms is based on the 12-6 LJ interaction potential model[59]. The corresponding LJ interaction potential parameters are 16.004 kcal·mol⁻¹ and 2.47 Å. In the simulation, each wall atom is connected to its original position through a harmonic spring, that is, $U = K [r(t) − r(0)]^2$, where $r(0)$ is the initial position of the platinum atom, $r(t)$ is the position of the platinum atom at $t$, and $K = 13 eVÅ^{-3}$ is the spring coefficient. The NVT ensemble is used to control the temperature of the simulation system. The relaxation temperature of the nanodroplet is 300 K. The Verlet velocity method is used to integrate the motion equation. The calculation time step is 10 fs, and the system equilibrium calculation is 1 ns. The quenching method is used to cool down to 130–250 K, while monitoring the changes in the potential energy and structure of the nanodroplets, and calculating until the nanodroplets nucleate and crystallize to achieve the liquid-solid phase transition of the nanodroplets. The mean square displacement (MSD) at time $t$ is calculated by Eq. (5):

$$MSD(t) = \frac{1}{n}\sum_{i=0}^{n}(r_i^t - r_i^0)^2 \quad (5)$$

where $r_i^t$ and $r_i^0$ are the positions of atom $i$ at time $t$ and 0, respectively.

In the NVT ensemble, pressure regulation is achieved by randomly adding different amounts of nitrogen molecules. A coarse-grained model is used for the nitrogen molecules in the system. Due to the limited size of the system, the number of nitrogen molecules is set to 20, 50, 100, 200 and 300 representing approximately 1.0, 2.5, 5.5, 11.0 and 17.5 atmospheres, respectively. The 12-6 LJ interaction potential model is used for the interaction between nitrogen molecules. The relevant interaction potential parameters are 0.0741 kcal·mol⁻¹ and 3.31 Å. The relaxation temperature of the system is 300 K and the quenching temperature is 205 K. The rest of the setup conditions are the same as above without the addition of nitrogen molecules.

### Identification of ice

To identify whether there is ice phase in supercooled water, the bond order parameters are used for quantitative discrimination based on the identification method proposed by Steinhardt et al.[60] and developed by Wolde et al.[61].

$$q_{lm}(i) = \frac{1}{N_b(i)}\sum_{j=1}^{N_b(i)} Y_{lm}\left(\mathbf{r}_{ij}\right) \quad (6)$$

$$c(i,j) = \frac{\sum_{m=-l}^{l} q_{lm}(i)q_{lm}^*(j)}{\left(\sum_{m=-l}^{l} q_{lm}(i)q_{lm}^*(i)\right)^{1/2}\left(\sum_{m=-l}^{l} q_{lm}(j)q_{lm}^*(j)\right)^{1/2}} \quad (7)$$

where $N_b(i)$ is the number of nearest neighbors of particle $i$, $Y_{lm}(\mathbf{r}_{ij})$ is a spherical harmonic function, $\mathbf{r}_{ij}$ is the unit vector connecting particle $i$ and one of the nearest neighbors of particle $j$, and * denotes the complex conjugate function. The $c(i, j)$ with a correlation to $q_3$ is used to distinguish crystals from liquid. The number of staggered bonds [$c(i, j) \leq -0.8$] and eclipsed bonds [$-0.35 \leq c(i,j) \leq 0.25$] are used to identify cubic ice, hexagonal ice and liquid water[62].

### Ice adhesion strength calculation of 2D and 3D coexisting ice

Based on the LAMMPS molecular simulation platform, 3D ice, 2D and 3D coexisting ice obtained from liquid-solid phase transition (temperature influence section) were used as initial configurations to study the molecular mechanism of de-icing. The simulation system adopts NVT ensemble temperature control. The ice relaxation temperatures

are 180, 205 and 230 K. The velocity Verlet method is used to integrate the motion equation. The calculation time step is 2 fs, and the system equilibrium calculation is 0.50 ns. Then, the acceleration field is applied to all ice atoms in the +$z$ direction, and its magnitude increases linearly with time $t$. The initial value of acceleration is 0 and increases linearly with $5.734*10^{-9}$ nm·fs⁻². A pulling force is applied to the ice, and the base reacts to the ice. The force $\mathbf{F}_A$ is monitored by collecting the force of the ice on the base.

### Gas-solid phase transition: molecular dynamics simulation of ice nucleation

Supplementary Fig. 19 shows a gas-liquid-solid three-phase system established by the LAMMPS molecular simulation platform. The system contains 5197 mW water molecules and the solid wall contains 8000 Pt atoms. The simulation system mainly includes high temperature wall (heat source), high temperature liquid layer, and low temperature wall (cold source). The sizes of nanochannels in three directions of $xyz$ are $15.69 \times 3.92 \times 313.76$ nm. The Pt-FCC structure is used on the high and low temperature solid walls, and the $x$, $y$ and $z$ directions are periodic boundary conditions. The interaction force between the water molecules in the liquid layer is described by mW model, and the interaction force between the wall and the water molecules is described by the 12-6 LJ interaction potential. Changing the interaction strength between the nanodroplet and the solid wall, $\varepsilon_{w-Pt}$ can realize the regulation of the wall from superhydrophobic to superhydrophilic in the range of 0.1–1.0 kcal·mol⁻¹. Gas-solid phase transition surface properties are the same as liquid-solid phase transition surface properties. The simulation system uses the NVT ensemble to control the temperature, and its relaxation temperature is 500 K. The velocity Verlet method is used to integrate the motion equation. The calculation time step is 10 fs, and the system equilibrium calculation is 1 ns. Then the NVE ensemble is used for the water molecules. Finally, the solid surface temperature is reduced to the target temperature by quenching. At the same time, the change of potential energy and liquid structure is monitored, and the calculation is carried out until crystallization, so as to obtain the gas-solid phase transition.

### Reporting summary

Further information on research design is available in the Nature Portfolio Reporting Summary linked to this article.

## Data availability

The data required to reproduce the key findings of this work are available in this article[63]. More detailed data are available from the corresponding authors upon request. Source data are provided with this paper[64].

## Code availability

The MD simulations are carried out by LAMMPS software package[65]. The input scripts and codes to reproduce the key findings of this work are available in this article[63]. More detailed data are available from the corresponding authors upon request.

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

## Acknowledgements
Y.M.L. is supported by the National Natural Science Foundation of China (No. U2268215) and the Key Research Program of Frontier Sciences of Chinese Academy of Sciences (No. QYZDY-SSW-DQC015). M.Y.Z. acknowledges the support by the National Natural Science Foundation of China (No. 41825015).

## Author contributions
J.J. and Y.L. designed the research, performed the research. J.J. conceived the idea, analyzed data and wrote-revised the paper. Y.L., D.S., G.T. and M.Z. wrote-reviewed and revised the paper. D.N. and F.Y. wrote the paper. All authors discussed the results and commented on the manuscript.

## Competing interests
The authors declare no competing interests.
