## [Peer Review File · Nature Communications]

Two-dimensional bilayer ice in coexistence with three-dimensional ice without confinementREVIEWER COMMENTS

Reviewer #1 (Remarks to the Author):

The authors conducted a molecular dynamic simulation study that demonstrated the coexistence of two-dimensional bilayer ice and three-dimensional ice without confinement. They found that above a critical value of the surface energy parameter, this coexistence occurs. The study explored the coexistence of ice during liquid-solid and gas-solid phase transitions. The authors discussed the deicing mechanism for 2D-3D coexisting ice, which has lower ice adhesion strength than 3D ice. However, the mechanism that leads to the coexistence of 2D-3D with the increase in surface energy parameters showing low ice adhesion strength needs further clarification.

Comments and questions:

1. What is the novelty and motivation of the work? Previous studies have reported the formation of 2D ice without nanoscale confinement including experimental work with molecular simulations showing practical scenarios (Ref. 36, 42-44). Figure 1 depicts the potential energy variation with an increase in energy parameters, which exhibits a trend similar to that reported previously for 2D ice formation (Ref. 44). What is the research gap? How would the present work provide new knowledge to the field?
2. What are the practical scenarios or conditions in which 2D-3D coexisting ice can form without confinement? How can the authors justify the validity of their study, given that it is purely a molecular simulation study?
3. What mechanism leads to the coexistence of 2D-3D ice with an increase in the surface energy parameter (Fig. 2)? The change in surface wettability from superhydrophobic to superhydrophilic has been well studied with variations in the surface energy parameter (Ref. 44). Why does the potential energy increase with temperature (Fig. S3)? The authors should clarify the reasons for the results mentioned in the present work.
4. Figure 3 shows that 3D ice has higher ice adhesion strength than 2D-3D coexisting ice. What is the reason for the lower ice adhesion strength for 2D-3D coexisting ice? The 2D-3D coexisting ice has a large contact area with a solid substrate which could provide a higher adhesion force. Moreover, the authors mention, "Fig. 3(a) and (b) reveal that the ice adhesion force of 2-dimensional ice in coexistence with 3-dimensional ice is significantly larger than that of 3-dimensional ice". The explanation contradicts the results shown. The authors should clarify this discrepancy.
5. The phase diagram for the gas-solid phase transition (Figure 4c) shows that 3D ice exists for energy parameters from 0.6-1 kcal/mol at low temperatures (< 210 K). However, the phase diagram for the liquid-solid phase transition (Figure 2d) shows the coexistence of 2D-3D ice for higher surface energy parameters and similar temperature conditions. Given that the gas-solid phase transition involves gas-liquid and liquid-solid transitions, why does 3D ice occur during the gas-solid phase transition for the higher value of the surface energy parameter?

Reviewer #2 (Remarks to the Author):

In the manuscript "Two-dimensional bilayer ice in coexistence with three-dimensional ice without confinement," the authors explore the possibility of controlling the formation of low- and high-dimensional ices by tuning the interactions of a water droplet on a surface. Understanding the conditions under which 2D ices can be produced is an active research topic that still needs to be further explored. In this regard, the authors offer an alternative perspective that differs from the usual ones based on molecular confinement. The fact that changing wettability results in a different ice growth mechanism may have important consequences for our understanding of water/ice behavior, as well as for many applications. Indeed, the results presented in the work open a debate about whether it is possible to produce and control the growth of other ice phases through this method.

The manuscript is overall quite convincing. The computational level is well tested and accurate for the results presented. Indeed, the authors have already published an article where a similar methodology was used with reliable results. The way different parameters affect their simulations is also well supported in the ESI file. Presenting wettability-temperature phase diagrams offers a way to probe their results, which may stimulate the experimental community. I think there is excellent ground for a Nat. Comm. article, but also that some complements and clarifications are needed.

1. The importance of the work should be addressed in more detail in the introduction. A nice background on confinement studies on 2D ices is presented as well as some results without confinement. However, the topic of 2D/3D coexistence and the novelty of their work is barely discussed. An explanation of the potential implications might increase the visibility of their work.
2. During the result discussion, citations should be added to reinforce some of the conclusions the authors present. For instance, it is said in both the liquid-solid and gas-solid analysis that "thermodynamic analyses show that the tendency to form 2-dimensional ice without nanoscale confinement is that appropriate water-surface interactions can compensate for the entropy loss in the freezing transition process." Which thermodynamic analysis? The authors do not present these results in the manuscript.
3. Regarding the simulations, the systems consist of a drop on a surface. Being a curved surface, there is a pressure difference inside the drop and outside, according to the Young-Laplace equation, which affects thermodynamic conditions such as vapor pressure. How is this phenomenon included in the model, and how does it affect their stability? Likewise, for a curved surface the Gibbs-Thomson equation predicts a melting point depression. Please provide some estimation of the changes in the vapor pressure and melting temperature.
4. The authors use an NVT ensemble in their simulations, however, nucleation of other competing ice phases is produced when considering different pressure conditions. Can their studies be extended to other pressure conditions?
5. In their previous work (ref. 44 of the manuscript. Proc. Natl. Acad. Sci. U.S.A. 116, 16723-16728 (2019)), the authors comment that using NpxyT simulations, the number of pentagonal, heptagonal, and octagonal rings in the interior part of the bilayer ice decreases and even goes to zero at longer simulation times. How does this effect affect the coexistence of 2D/3D ice? Since the structure of the 2D ice changes, could the adhesion energy of the 2D ice (de-icing mechanism) be modified? This point should be addressed in the manuscript as it might have implications in the conclusions of the work.
6. Unequivocally distinguishing ice types and water local structure is crucial in describing nucleation phenomena. The performance of a given order parameter has been widely discussed in the literature, since it may yield different results. (J. Chem. Phys. 144, 034501 (2016)) How do different order parameters affect the formation of 2D and 3D ice in their simulations? Have the authors used a minimum mislabeling procedure to determine the values used for the q_6 and q_3 parameters? The influence of these parameters on the results presented in the manuscript should be stated, if any.
7. I believe that the manuscript would benefit from a discussion on how the simulations can be translated into experiments. For instance, mentioning that changes in the ϵ parameter could potentially be achieved through modifications of lattice planes or surface characteristics would make the paper more accessible to experimentalists and broaden its audience.

Minor comments:

1. There are some sentences that are quite similar in the solid-gas and solid-liquid discussion. It would be beneficial for the authors to unify the text.
2. In Figure 1 (left), the red axis is not explained in either the text or the caption. Please include an explanation.
3. The citation format needs to be revised and unified.

Reviewer #3 (Remarks to the Author):

The authors are using computer simulations to observe the solidification process of water on solid surfaces.

While I think their point of view is interesting, they do not seem to have investigated it in sufficient depth. They give various descriptions of the crystallization process, but they are not accompanied by data to corroborate them, so we can only guess them from the context. By the time I got to page 5, I was confronted with various points of uncertainty. I think this study (or this paper) is incomplete.

Line 94: "The potential energy per water molecule at 250 K is significantly higher than that of per water molecule at other temperatures. ": This sentence is unclear; it mentions 250 K, but without any data, I have no idea what they are referring to.

Line 109 Figure 1(b) The scale on the right has no explanation and makes no sense.

Regarding the plot of 0.5 kcal/mol in red dots, the authors say it shows that liquid and solid coexist, but if they coexist, wouldn't the graph reflect only the liquid character with a large diffusion coefficient?

Around line 124: various explanations are given, but no evidence seems to be presented. For example,

Line 122: They state that nucleation occurs in the droplet and not on the solid surface because the surface is superhydrophobic, but they do not show a causal relationship between the two, which reads as just an unfounded prediction.

Line 128: "With increasing wettability, the contact angle of nanodroplet decreases and the contact radius increases, which enhances the interaction between droplet and the surface." I think the causal relationship is the opposite.

Line 131: "Nanodroplet on surface with higher wettability is more likely to nucleate and crystallize due to the very high adsorption energy of the surface" It is just a guess and no evidence is provided.

Line 134: I don't know what a "large-scale hydrogen bond" is.

Line 135: The phenomenon of molecules aligning to form crystals is not limited to water. I don't think water is particularly prone to ordering, but if you claim that, you need evidence.

Line 136: It is an obvious statement over 10 lines.

Line 143: This is not true. The most stable phase is the phase with the lowest free energy, not the lowest energy.

Line 145 is correct, but it is an obvious thermodynamic fact and I am not sure what they are trying to emphasize.

Line 147: "The nucleation and crystallization of nanodroplets mainly occur on the solid surface" Since no conditions are set and no evidence is given, it is not clear what kind of situation is envisioned in the statement.

Response Sheet

Title: Two-dimensional bilayer ice in coexistence with three-dimensional ice without confinement

Paper number: NCOMMS-23-05786

We are glad to see that all reviewers have recommended our manuscript for publication. We would like to thank the reviewers for their constructive suggestions. In the following, we have responded point-by-point to all their comments and questions and indicated the corresponding changes made in the revised manuscript. All the changes are marked in blue in the revised manuscript.

Reviewer #1 (Remarks to the Author):

The authors conducted a molecular dynamic simulation study that demonstrated the coexistence of two-dimensional bilayer ice and three-dimensional ice without confinement. They found that above a critical value of the surface energy parameter, this coexistence occurs. The study explored the coexistence of ice during liquid-solid and gas-solid phase transitions. The authors discussed the deicing mechanism for 2D-3D coexisting ice, which has lower ice adhesion strength than 3D ice. However, the mechanism that leads to the coexistence of 2D-3D with the increase in surface energy parameters showing low ice adhesion strength needs further clarification.

Reply: We appreciate these positive comments about our work. In the following, we address the reviewer's questions and indicate revisions made in the revised manuscript and Supplementary Information (SI) accordingly.

Comments and questions:

1. What is the novelty and motivation of the work? Previous studies have reported the formation of 2D ice without nanoscale confinement including experimental work with

molecular simulations showing practical scenarios (Ref. 36, 42-44). Figure 1 depicts the potential energy variation with an increase in energy parameters, which exhibits a trend similar to that reported previously for 2D ice formation (Ref. 44). What is the research gap? How would the present work provide new knowledge to the field?

Reply: Thank you for your constructive question.

- a) The knowledge on 2D-3D coexisting ice formation and de-icing characteristics at molecular level remains blank. The influences of the surface wettability, temperature, pressure and phase transition mode on the formation of 2D-3D coexisting ice without confinement has not been studied. Our findings fill the gap about the coexistence of 2D bilayer ice and 3D ice without nanoscale confinement. We explore the possibility of controlling the formation of low- and high-dimensional ices by tuning the interactions of a water droplet on a surface. Here, we show a groundbreaking work about the formation and growth of 2D-3D coexisting ice and de-icing characteristics in which the phase diagrams about temperature and wettability are predicted for liquid-solid and gas-solid phase transitions. Changing wettability leads to different ice growth mechanisms, which helps us to better understand water-ice behavior. This study deepens our understanding of 2D-3D coexisting ice without confinement, and is a step forward towards a predictive model for water-ice phase developments which can be used to guide and interpret future experiments. Indeed, the results presented in our work open the debate about whether it is possible to produce and control the growth of ice phases in experiments and reality.
- b) We used the same method, i.e. by tuning the interactions of a water droplet on a surface, to determine a critical value of the surface energy parameter characterizing the liquid-solid interface interaction. Above this critical value, 2D and 3D coexisting ice can be formed on the surface. This finding is new and significant for exploring the possibility of controlling the formation of low- and high-dimensional ices by tuning the interactions of a water droplet on a surface.
- c) Our study has elucidated the physical mechanism behind the formation of 2D-3D coexisting ice, which offers further insight into how the formation of 2D-3D

coexisting ice is related to the water-surface interactions. Our findings provide a molecular-level explanation of 2D-3D coexisting ice on surfaces without confinement, thereby offering a guide for growing 2D-3D coexisting ice on different solid surfaces. Our work will hopefully motivate future experimental studies of the formation of low- and high-dimensional ice on a variety of realistic surfaces. 2D ice on the surface can promote or inhibit the formation of 3D ice, which has potential application value for the design and development of anti-icing materials.

The updated content above have been incorporated to the revised manuscript on lines 68-83, 468-475.

2. What are the practical scenarios or conditions in which 2D-3D coexisting ice can form without confinement? How can the authors justify the validity of their study, given that it is purely a molecular simulation study?

Reply: Thank you for your question.

- a) The growth of 2D ice has been widely observed experimentally on graphene [1], Au(111) surface [2], Ru(0001) substrate and Pt(111) surface [3] at low temperature without confinement. For graphene surface, thin films of water were deposited on the substrate at temperatures ranging from 20-152K. The 2D ice was only observed for 100-135 K, while 3D ice was observed at $T > 140\text{K}$. For Au(111) surface, the 2D ice was observed for $T = 120\text{K}$. For comparison, the 2D ice can be observed on the Pt(111) surface and Ru(0001) substrate for $T < 140\text{K}$. Hence, the experimental results for Pt(111) surface are consistent with our predictions for $130\text{K} \leq T < 140\text{K}$, that is, there is 2D ice. By adjusting the droplet size and surface wettability, the practical working conditions of 2D-3D coexisting ice can be obtained by experiment. The practical conditions for 2D-3D coexisting ice without confinement are estimated as surface energy parameter $0.43 \text{ kcal/mol} < \epsilon_{\text{w-pt}} \leq 1.0 \text{ kcal/mol}$ and temperature range $130\text{K} \leq T < 140\text{K}$. The contact angle of the water droplet on the platinum metal surface is $\theta < 75^\circ$, which is in the hydrophilic and superhydrophilic state. It is

important to obtain the hydrophilic and superhydrophilic surface by constructing rough structure and adjusting wettability. There are two main ways to prepare hydrophilic and superhydrophilic surface. One is light-induced method, which uses the photocatalytic process of photocatalytic materials such as TiO₂ films to produce hydrophilicity and superhydrophilicity. The second is to construct microscopic rough structures on the hydrophilic surface to make it more hydrophilic, by increasing the roughness of the substrate surface.

- b) For Pt(111) surface, 2D ice can be observed in the experimental temperature range of $T < 140\text{K}$ [3]. For $130\text{K} \leq T < 140\text{K}$, the experimental results are already consistent with our predictions for liquid-solid phase transition by MD simulation. In addition, the intensity of the right low peak is significantly greater than that of the left low peak as shown in Fig. 2b, revealing that the growth of the 2D double ice begins at the bottom, which is consistent with the results in Reference [40]. The 3D ice is first detached from the surface at the right edge, becoming increasingly tilted relative to the substrate and eventually detached from the substrate. This angular ice detachment mechanism is consistent with the experimental observation results [55]. Actually, we have validated the molecular simulation study in our previous work [4] (3.1. *Validation of the simulation method*). To further verify the water model and the simulation setup, a homogeneous ice nucleation simulation is performed and its nucleation rate is estimated as $1.92 \pm 0.12 \times 10^{32} \text{ m}^{-3} \text{ s}^{-1}$, which is comparable to the order of magnitude of the simulation results in Ref. [26]). In addition, we can find similar methods from previous studies and five cited references [6-10] below are listed as examples,

[1] Kimmel, G.A. et al. No confinement needed: Observation of a metastable hydrophobic wetting two-layer ice on graphene. *J. Am. Chem. Soc.* 131, 12838-12844 (2009).

[2] Ma, R. et al. Atomic imaging of edge structure and growth of a two-dimensional hexagonal ice. *Nature* 577, 60-63 (2020).

[3] Maier, S., Lechner, B.A., Somorjai, G.A. & Salmeron, M. Growth and structure of the first layers of ice on Ru(0001) and Pt(111). *J. Am. Chem. Soc.* 138, 3145-3151 (2016).

[4] Jiang J., Li G.X., Sheng Q. & Tang G.H. Microscopic mechanism of ice nucleation: The effects of surface rough structure and wettability. *Appl. Surf. Sci.* 510, 145520 (2020).

- [5] Matsumoto, M., Saito, S. & Ohmine, I. Molecular dynamics simulation of the ice nucleation and growth process leading to water freezing. *Nature* 416, 409-413 (2002).
- [6] Moore, E.B. & Molinero, V. Structural transformation in supercooled water controls the crystallization rate of ice. *Nature* 479, 506-509 (2011).
- [7] Lupi, L. et al. Role of stacking disorder in ice nucleation. *Nature* 551, 218-222 (2017).
- [8] Zhu, C.Q. et al. Direct observation of 2-dimensional ices on different surfaces near room temperature without confinement. *Proc. Natl. Acad. Sci. USA* 116, 16723-16728 (2019).
- [9] Bi, Y.F., Cao, B.X. & Li, T.S. Enhanced heterogeneous ice nucleation by special surface geometry. *Nat. Commun.* 8, 15372 (2017).

The updated content have been incorporated to the revised manuscript on lines 199-202 and 308-310.

3. What mechanism leads to the coexistence of 2D-3D ice with an increase in the surface energy parameter (Fig. 2)? The change in surface wettability from superhydrophobic to superhydrophilic has been well studied with variations in the surface energy parameter (Ref. 44). Why does the potential energy increase with temperature (Fig. S3)? The authors should clarify the reasons for the results mentioned in the present work.

Reply: Thank you for your constructive question.

- a) The coexistence of 2D-3D ice start to appear as the surface energy parameter increases over a critical value, and the specific critical value depends on the surface wettability and temperature. The mechanism for the 2D-3D coexisting ice without nanoscale confinement is attributed to overcoming the free energy barrier and the satiation of the Bernal-Fowler ice rules and the appropriate water-surface interaction, which can compensate for the entropy loss caused by the freezing transition process. For the liquid-solid phase transition part, the purpose of our investigation is to predict the phase diagram of temperature and energy parameters to distinguish liquid water, 2D double-layer ice and 3D ice. In addition, 2D-3D coexisting ice is also obtained. However, Ref. 42 studied 2D ice formation on various surfaces and the dependence of the 2D crystalline structure on the hydrophobicity and morphology of the underlying surface.

- b) In the NVT ensemble, the volume of the simulation system box is constant. As the temperature increases, the pressure of the system increases, so does the potential energy of water molecules.

The updated content above have been incorporated to the revised manuscript on lines 224-227 and 101-104.

4. Figure 3 shows that 3D ice has higher ice adhesion strength than 2D-3D coexisting ice. What is the reason for the lower ice adhesion strength for 2D-3D coexisting ice? The 2D-3D coexisting ice has a large contact area with a solid substrate which could provide a higher adhesion force. Moreover, the authors mention, “Fig. 3(a) and (b) reveal that the ice adhesion force of 2-dimensional ice in coexistence with 3-dimensional ice is significantly larger than that of 3-dimensional ice”. The explanation contradicts the results shown. The authors should clarify this discrepancy.

Reply: Thank you for your constructive question.

- a) Ice adhesion strength is the ice adhesion force divided by the ice-solid substrate contact area ($\sigma=F/A$). Ice adhesion strength depends on the ice adhesion force and the ice contact area. The contact area of 2D-3D coexisting ice and solid substrate is 2.35 times that of 3D ice and solid substrate ($A_{2D-3D}/A_{3D}=2.35$). The 2D-3D coexisting ice has a large contact area with a solid substrate, which could provide a higher adhesion force. However, the adhesion force of 2D-3D coexisting ice is no more than 2 times that of 3D ice ($F_{2D-3D}/F_{3D}<2$). Therefore, ice adhesion strength for 2D-3D coexisting ice is lower than that of 3D ice.

$$\sigma_{3D}/\sigma_{2D-3D} = (F_{3D}/A_{3D})/(F_{2D-3D}/A_{2D-3D}) = 2.35 F_{3D}/F_{2D-3D} > 1$$

- b) Apologies for the writing mistake. We have revised the sentence to “Fig. 4a reveals that the ice adhesion force of 2D ice in coexistence with 3D ice is significantly larger than that of 3D ice. Fig. 4b shows the adhesion strength of 2D-3D coexisting ice is smaller than that of 3D ice, which is attributed to the combined effect of ice adhesion force and ice contact area.”

The updated content above have been incorporated to the revised manuscript on lines 292-302.

5. The phase diagram for the gas-solid phase transition (Figure 4c) shows that 3D ice exists for energy parameters from 0.6-1 kcal/mol at low temperatures (< 210 K). However, the phase diagram for the liquid-solid phase transition (Figure 2d) shows the coexistence of 2D-3D ice for higher surface energy parameters and similar temperature conditions. Given that the gas-solid phase transition involves gas-liquid and liquid-solid transitions, why does 3D ice occur during the gas-solid phase transition for the higher value of the surface energy parameter?

Reply: Thank you for your constructive question.

For $200 \text{ K} \leq T \leq 210 \text{ K}$ and $0.6 \leq \varepsilon \leq 1.0$ kcal/mol, gas-solid phase transition shows that 3D ice exists, and the liquid-solid phase transition shows the presence of 2D-3D coexisting ice. The difference is mainly attributed to the phase change form, supercooled degree and droplet shape before freezing. For the gas-solid phase transition, the solid surface temperature is reduced from 500 K to the target temperature by quenching. The gas-solid phase transition involves gas-liquid and liquid-solid transitions. The water vapour first condenses into film water and then freezes in the form of 3D ice. For the liquid-solid phase transition, the nanodroplets are quenched from 300K to the target temperature. The water molecules close to the wall in the nanodroplets grow rapidly to form 2D ice, and the remaining water molecules exist in the form of dropwise-water. When the critical nucleus is formed by overcoming the free energy barrier, the 3D ice nucleus grows spontaneously and crystallizes. Therefore, 2D ice and 3D ice coexist.

Reviewer #2 (Remarks to the Author):

In the manuscript "Two-dimensional bilayer ice in coexistence with three-dimensional ice without confinement," the authors explore the possibility of controlling the formation of low- and high-dimensional ices by tuning the interactions of a water droplet on a surface. Understanding the conditions under which 2D ices can be produced is an active research topic that still needs to be further explored. In this regard, the authors offer an alternative perspective that differs from the usual ones based on molecular confinement. The fact that changing wettability results in a different ice growth mechanism may have important consequences for our understanding of water/ice behavior, as well as for many applications. Indeed, the results presented in the work open a debate about whether it is possible to produce and control the growth of other ice phases through this method.

The manuscript is overall quite convincing. The computational level is well tested and accurate for the results presented. Indeed, the authors have already published an article where a similar methodology was used with reliable results. The way different parameters affect their simulations is also well supported in the ESI file. Presenting wettability-temperature phase diagrams offers a way to probe their results, which may stimulate the experimental community. I think there is excellent ground for a Nat. Comm. article, but also that some complements and clarifications are needed.

Reply: We would like to thank the reviewer for their very positive and constructive comments. In the following, we address the reviewer's questions and indicate revisions made in the revised manuscript and Supplementary Information (SI) accordingly.

1. The importance of the work should be addressed in more detail in the introduction. A nice background on confinement studies on 2D ices is presented as well as some results without confinement. However, the topic of 2D/3D coexistence and the novelty of their work is barely discussed. An explanation of the potential implications might increase the visibility of their work.

Reply: Thank you for your constructive question.

Previous investigations have significantly deepened our understanding of 2D ice formation without confinement, while the 2D-3D coexisting ice formation and de-icing characteristics at molecular level remains blank. The influence of the surface wettability, temperature, pressure and phase transition mode on the formation of 2D-3D coexisting ice without confinement has not been systematically studied. Our novel findings fill the gap where 2D bilayer ice coexists with 3D ice without nanoscale confinement. We explore the possibility of controlling the formation of low- and high-dimensional ices by tuning the interactions of a water droplet on a surface. Here, we report a groundbreaking work about the formation and growth of 2D-3D coexisting ice and de-icing characteristics in which the phase diagrams about temperature and wettability are predicted for liquid-solid and gas-solid phase transitions. Changing wettability leads to different ice growth mechanisms, which helps us to better understand water/ice behavior. This study deepens our understanding of 2D-3D coexisting ice without confinement and is a step forward towards a predictive model for the formation and growth of other ice phases. Such a model can help to guide and interpret future experiments. Indeed, the results presented in our work open a debate about whether it is possible to produce and control the growth of other ice phases through our method. Physical mechanism related to the formation of 2D-3D coexisting ice is elucidated, which offers insight into how the 2D-3D coexisting ice formation is related to the water-surface interactions. Our findings provide a molecular-level understanding of 2D ice formation on surfaces without confinement, thereby offering a guide for growing 2D ices on different solid surfaces. The formation of 2D-3D coexisting ice without confinement revealed in this work will hopefully motivate future experimental studies on a variety of realistic surfaces. 2D ice has an important influence on the growth of 3D ice. If there is a 2D ice, the 3D ice will attach and grow on the surface and be very stable. However, if there is no 2D ice, the formed 3D ice has a small contact surface with the surface and is easily blown

away by the wind. Therefore, we can design and develop anti-icing materials more specifically according to the structure of 2D and 3D coexisting ice.

The updated content above have been incorporated to the revised manuscript on lines 68-83, 468-475.

2. During the result discussion, citations should be added to reinforce some of the conclusions the authors present. For instance, it is said in both the liquid-solid and gas-solid analysis that "thermodynamic analyses show that the tendency to form 2-dimensional ice without nanoscale confinement is that appropriate water-surface interactions can compensate for the entropy loss in the freezing transition process." Which thermodynamic analysis? The authors do not present these results in the manuscript.

Reply: Thank you for your kind comments. We have cited the relevant reference [42] to reinforce the conclusions.

The updated content have been incorporated to the revised manuscript on lines 177-179 and 401-403.

3. Regarding the simulations, the systems consist of a drop on a surface. Being a curved surface, there is a pressure difference inside the drop and outside, according to the Young-Laplace equation, which affects thermodynamic conditions such as vapor pressure. How is this phenomenon included in the model, and how does it affect their stability? Likewise, for a curved surface the Gibbs-Thomson equation predicts a melting point depression. Please provide some estimation of the changes in the vapor pressure and melting temperature.

Reply: Thank you for your kind comments.

- a) Pressure difference inside the drop and outside is indeed an existing phenomenon especially for a nanodroplet based on the Young-Laplace equation. Firstly, the vapor pressure mentioned above is almost independent of the droplet radius based on previous research result [1]. If the effect of droplet size is considered, it should have a higher vapor pressure than the flat liquid surface.

However, the vapour-pressure independence of droplet size may originate from system setup in MD simulation. Considering the mean free path of vapor molecules, the rarefaction effect will appear in such a simulation system. It should be noted that for a droplet system in MD simulation the rarefaction effect is almost unavoidable due to the limited size of the simulation system. Therefore, combining the Young-Laplace equation and rarefaction effect, the vapour-pressure independence of droplet size may appear. Of course, current research has not yet clearly verified this speculation, which should be an interesting topic for future research. Here, we just provide a reasonable speculation based on previous research results.

- b) The melting point depression for a particle with a certain radius of curvature could be described using the Gibbs-Thomson equation. However, the original Gibbs-Thomson equation may not be suitable for this simulation size, according to a previous study [2]. At least one correction for surface tension related to the droplet size is required to accurately describe the melting point depression. Furthermore, in our simulation, the decrease in the melting point does not affect the ice progress. For example, we set a lower icing temperature in which the ice structure does not show any differences from the results at the current temperature. However, as mentioned by the reviewer, the Gibbs-Thomson equation is quite important for the prediction of melting point. But, this is not the focus of this study in which a lower temperature ensuring the occurrence of the freezing process is sufficient.

[1] Langroudi S M M, Ghassemi M, Shahabi A, et al. A molecular dynamics study of effective parameters on nano-droplet surface tension. *Journal of Molecular Liquids*, 2011, 161(2): 85-90.

[2] Wu N, Lu X, An R, et al. Thermodynamic analysis and modification of Gibbs–Thomson equation for melting point depression of metal nanoparticles. *Chinese Journal of Chemical Engineering*, 2021, 31: 198-205.

4. The authors use an NVT ensemble in their simulations, however, nucleation of other competing ice phases is produced when considering different pressure conditions. Can their studies be extended to other pressure conditions?

Reply: Thank you for your constructive question. We have studied the influence mechanism of pressure on coexisting ice in second section, as follows,

Liquid-solid phase transition: the effect of pressure on 2D ice in coexistence with 3D ice without confinement

In order to investigate the influence of pressure on coexisting ice, five pressure conditions were chosen for molecular dynamic simulation at 205K. In the NVT ensemble, pressure regulation is achieved by adding different numbers of nitrogen molecules (Supplementary Fig. 9). Due to the limited size of the system box, 20, 50, 100, 200 and 300 nitrogen molecules were added to obtain approximate pressures of 1.0, 2.5, 5.5, 11.0 and 17.5 atmospheres, respectively. Fig. 3a shows that the contact angle of nanodroplets decreases with increasing gas pressure, and the droplet curvature decreases first and then increases with increasing gas pressure for energy parameter $\varepsilon=0.4\text{kcal/mol}$. Taking pressure of 11.0 atm as an example, for $0.1\text{ kcal/mol}\leq\varepsilon<0.43\text{kcal/mol}$, the liquid-solid phase transition only generates 3D ice with disordered stacking of hexagonal ice and cubic ice (Supplementary Fig. 10). For $0.43\text{ kcal/mol}\leq\varepsilon\leq1.0\text{ kcal/mol}$, the liquid-solid phase transition produces coexisting ice, which is composed of 2D ice and 3D ice stacked disorderly with hexagonal ice and cubic ice (Supplementary Fig. 11). With increasing wetting characteristics, the contact angle of droplets decreases, and droplets gradually spread on the wall. After quenching, 2D ice begins to form instantaneously, and 3D ice is formed after overcoming the nucleation energy barrier. The variation of potential energy with time also reflects the occurrence of nucleation and icing process (Supplementary Fig. 12). When the surface energy parameters is 0.5 kcal/mol, 2D ice grows and merges slowly for pressure 11.0 atm (and 17.5 atm) as shown in Fig. 3b, which is attributed to the combined effect of surface wettability and gas pressure. After calculation of 200ns, the 2D ice is still composed of double-layer 5-, 6-, 7-membered ring water molecules.

Fig. 3 Effect of gas pressure on growth characteristics of 2D-3D ice. **a** Variation of contact angle and curvature versus pressure. **b** A sequence of snapshots of the coexisting ice growth process. **c** Phase diagram of 3D ice and 2D-3D coexisting ice with respect to gas pressure and energy parameter ε .

The phase diagram about energy parameters and pressure is predicted to distinguish 2D ice, 3D and 2D-3D coexisting ice by a large number of molecular dynamic simulations. The critical value of the surface energy parameter is estimated to be 0.43 kcal/mol, which is the same as that of the nitrogen-free system, indicating that the formation of 2D-3D coexisting ice on the platinum surface is not very sensitive to the applied gas pressure conditions. The effects of multiple pressure conditions and multiple surface wetting characteristics on the liquid-solid phase transition of water molecules are revealed as shown in Fig. 3c. For $1.0 \text{ atm} \leq P \leq 17.5 \text{ atm}$ and $\varepsilon < 0.43 \text{ kcal/mol}$, the supercooled droplet undergoes liquid-solid phase transition to solid phase, and the 3D ice is composed of disordered hexagonal ice and cubic ice. For $1.0 \text{ atm} \leq P \leq 17.5 \text{ atm}$ and $0.43 \text{ kcal/mol} \leq \varepsilon \leq 1.0 \text{ kcal/mol}$, liquid-solid phase transition occurs, and the coexisting ice consists of 2D ice and 3D hexagonal ice and cubic ice. For $0.43 \text{ kcal/mol} \leq \varepsilon < 0.8 \text{ kcal/mol}$, the 2D ice in coexisting ice is composed of double-layer 5-, 6-, 7-membered ring water molecules. For $0.8 \text{ kcal/mol} \leq \varepsilon \leq 1.0 \text{ kcal/mol}$, the 2D ice in coexisting ice is finally composed of a layer of 5-, 6-, 7-membered ring water molecules and a layer of 4-, 6-

membered ring water molecules near solid surface (Supplementary Fig. 13). The appearance of the 4-membered square ring water molecules is attributed to the surface being too hydrophilic and gas pressure.

The updated figure and the explanation above have been incorporated to the revised manuscript on lines 228-274, 501-508 and Supplementary Information on lines 85-117.

5. In their previous work (ref. 44 of the manuscript. Proc. Natl. Acad. Sci. U.S.A. 116, 16723-16728 (2019)), the authors comment that using NpxyT simulations, the number of pentagonal, heptagonal, and octagonal rings in the interior part of the bilayer ice decreases and even goes to zero at longer simulation times. How does this effect affect the coexistence of 2D/3D ice? Since the structure of the 2D ice changes, could the adhesion energy of the 2D ice (de-icing mechanism) be modified? This point should be addressed in the manuscript as it might have implications in the conclusions of the work.

Reply: Thank you for your constructive question. In the NVT ensemble, when the calculation time reaches 200 ns, the 2D and 3D coexisting ice structures are relatively stable. The 2D ice is still composed of double-layer 5-, 6-, 7-membered ring water molecules, and the proportion of 5-, 7-membered ring water molecules is basically unchanged, indicating that the 2D and 3D coexisting ice is not sensitive to the calculation time when it is sufficiently long (Supplementary Fig. 8). The ice adhesion force of the 2D and 3D coexisting ice is not affected by the calculation time.

Supplementary Fig. 8 2D-3D ice structure variation

In NPxyT ensemble, the pressure is set to $P_{xy}=1\text{atm}$. With the increase of calculation time, the 5-, 7-membered rings in the double-layer ice are reduced. After 200ns of calculation, the 2D ice composed of metastable double-layer 5-, 6-, 7-membered ring water molecules is transformed into steady-state 6-membered ring 2D ice. 3D ice structure is not sensitive to the calculation time.

Supplementary Fig. 17 2D and 3D coexisting ice structure variation at 205K

At 30ns, the 2D ice is composed of double-layer 5-, 6-, 7-membered ring water molecules. At 200ns, the 2D ice is composed of double-layer 6-membered ring water molecules. With the increase of energy parameter ϵ_{w-Pt} , the ice adhesion force at 200ns becomes gradually larger than that at 30 ns, which is attributed to the combined effect of surface wetting characteristics, 2D ice structure and ice freezing time.

Supplementary Fig. 18 Relationship between the ice adhesion force and wettability energy parameter ϵ_{w-Pt} for the coexisting ice at 205K

The updated figure and the explanation above have been incorporated to the revised manuscript on lines 195-199, 364-372 and Supplementary Information on lines 84-89, 134-148.

6. Unequivocally distinguishing ice types and water local structure is crucial in describing nucleation phenomena. The performance of a given order parameter has been widely discussed in the literature, since it may yield different results. (J. Chem. Phys. 144, 034501 (2016)) How do different order parameters affect the formation of 2D and 3D ice in their simulations? Have the authors used a minimum mislabeling procedure to determine the values used for the q6 and q3 parameters? The influence of these parameters on the results presented in the manuscript should be stated, if any.

Reply: Thank you for your comments.

- a) To identify whether there is ice phase in supercooled water, the bond order parameters are used for quantitative discrimination based on the identification method proposed by Steinhardt et al. [62] and developed by Wolde et al. [63].

$$q_{lm}(i) = \frac{1}{N_b(i)} \sum_{j=1}^{N_b(i)} Y_{lm}(\mathbf{r}_{ij}) \quad (5)$$

$$c(i, j) = \frac{\sum_{m=-l}^l q_{lm}(i) q_{lm}^*(j)}{\left(\sum_{m=-l}^l q_{lm}(i) q_{lm}^*(i)\right)^{1/2} \left(\sum_{m=-l}^l q_{lm}(j) q_{lm}^*(j)\right)^{1/2}} \quad (6)$$

where $N_b(i)$ is the number of nearest neighbors of particle i , $Y_{lm}(\mathbf{r}_{ij})$ is a spherical harmonic function, \mathbf{r}_{ij} is the unit vector connecting particle i and one of the nearest neighbors of particle j , and $*$ denotes the complex conjugate function. Adopting the CHILL+ algorithm, the $c(i, j)$ with a correlation to q_3 ($l=3$) is used to distinguish crystals (cubic and hexagonal ice) from liquid. The number of staggered bonds [$c(i, j) \leq -0.8$] and eclipsed bonds [$-0.35 \leq c(i, j) \leq 0.25$] are used to identify cubic ice, hexagonal ice and liquid water [64]. According to the reference (J. Chem. Phys. 144, 034501 (2016)), we calculate and analyze the effects of local order parameters q_3 ($l=3$), q_4 ($l=4$), q_6 ($l=6$) on liquid water, 2D ice and 3D ice on Pt surface. Fig.1 shows the distribution of the local order parameters q_3 , q_4 , q_6 for 3D ice, 2D ice and liquid water. The distributions of q_3 ($l=3$) for liquid water and 2D ice show a large overlap in Fig. 1a. The 3D ice can be distinguished from 2D ice and liquid water. The distributions of q_4 ($l=4$) for liquid water, 2D ice and 3D ice show a fairly large overlap in Fig. 1b. It is difficult to identify ice molecular from liquid water molecular. In contrast, the distributions of q_6 ($l=6$) for liquid water and 2D ice show a fairly large overlap. There is a small overlap in the distribution of local q_6 parameters between 2D ice and 3D ice. It is difficult to identify 2D ice from liquid water. Therefore, we choose CHILL+ algorithm (about q_3) to identify cubic ice, hexagonal ice and liquid water. In our simulation system, there are 2430 water molecular. After quenching and cooling, 2D ice grows close to the

wall, 3D ice growth is based on overcoming the free energy barrier. The proportion of 2D ice in coexisting ice is very small. The number of 2D ice molecules is obtained by direct observation based on reference [42]. Previous methods for identifying ice phase from liquid water are mostly based on bulk water and homogeneous nucleation. The identification of 2D ice depended on solid surface growth is a challenging subject, and we need to further explore the relevant local order parameters. It will be our next research direction.

Fig. 1 Distribution of the local order parameters a q_3 for 3D ice, 2D ice and liquid water. b q_4 for 3D ice, 2D ice and liquid water. c q_6 for 3D ice, 2D ice and liquid water.

- b) We are very sorry for not using a minimum mislabeling procedure to determine the bond order parameter values because we adopt the CHILL+ algorithm [64] related to bond order parameter.

The updated content above have been incorporated to the revised manuscript on lines 510-519.

7. I believe that the manuscript would benefit from a discussion on how the simulations can be translated into experiments. For instance, mentioning that changes in the ϵ parameter could potentially be achieved through modifications of lattice planes or surface characteristics would make the paper more accessible to experimentalists and broaden its audience.

Reply: Thank you for your kind comments. Our work will hopefully motivate future experimental studies of the formation of low- and high-dimensional ice on a variety of realistic surfaces. The change of the ϵ parameter could potentially be achieved through modifications of lattice planes or surface characteristics. The adjustment of surface characteristics is achieved by adjusting the wettability or

constructing rough structures. 2D ice on the surface can promote or inhibit the formation of 3D ice, which has potential application value for the design and development of anti-icing materials. The conditions for 2D-3D coexisting ice without confinement are surface energy parameter $0.43 \text{ kcal/mol} < \varepsilon_{\text{w-Pt}} \leq 1.0 \text{ kcal/mol}$. The contact angle of the water droplet on the platinum metal surface is $\theta < 75^\circ$, which is in the hydrophilic and superhydrophilic state. It is important to obtain the hydrophilic and superhydrophilic surface by constructing rough structure and adjusting wettability. There are two main ways to prepare hydrophilic and superhydrophilic surface. One is light-induced method, which uses the photocatalytic process of photocatalytic materials such as TiO_2 films to produce hydrophilicity and superhydrophilicity. The second is to construct microscopic rough structures on the hydrophilic surface to make it more hydrophilic, by increasing the roughness of the substrate surface.

The updated content above have been incorporated to the revised manuscript on lines 468-475.

Minor comments:

1. There are some sentences that are quite similar in the solid-gas and solid-liquid discussion. It would be beneficial for the authors to unify the text.

Reply: Thank you for your question. We have revised the section on solid-gas discussion accordingly.

2. In Figure 1 (left), the red axis is not explained in either the text or the caption. Please include an explanation.

Reply: Thank you for your question. We have added the sentence “The mean-square displacement of $\varepsilon=0.50 \text{ kcal/mol}$ corresponds to the vertical axis on the right-hand side” in the revised manuscript.

3. The citation format needs to be revised and unified.

Reply: Thank you for your question. We have revised and unified the citation format in the manuscript.

Reviewer #3 (Remarks to the Author):

The authors are using computer simulations to observe the solidification process of water on solid surfaces.

While I think their point of view is interesting, they do not seem to have investigated it in sufficient depth. They give various descriptions of the crystallization process, but they are not accompanied by data to corroborate them, so we can only guess them from the context. By the time I got to page 5, I was confronted with various points of uncertainty. I think this study (or this paper) is incomplete.

Reply: We sincerely thank the reviewer's positive comments and useful suggestions. In the following, we further clarify the reviewer's concerns and indicate revisions made in the revised manuscript and Supplementary Information (SI) accordingly.

1. Line 94: "The potential energy per water molecule at 250 K is significantly higher than that of per water molecule at other temperatures. ": This sentence is unclear; it mentions 250 K, but without any data, I have no idea what they are referring to.

Reply: Thank you for your question. The potential energy per water molecule at 250 K is significantly higher than that of per water molecule at other temperatures as shown in Supplementary Fig. 3.

Supplementary Fig. 3 Potential energy per water molecule vs temperature

2. Line 109 Figure 1(b): The scale on the right has no explanation and makes no sense. Regarding the plot of 0.5 kcal/mol in red dots, the authors say it shows that liquid and solid coexist, but if they coexist, wouldn't the graph reflect only the liquid character with a large diffusion coefficient?

Reply: Thank you for your constructive question, which help us improve the Fig.1b to the version below. In Fig.1b, the vertical axis on the right-hand side corresponds to the mean-square displacement of $\epsilon=0.50$ kcal/mol. The mean-square displacement of $\epsilon=0.50$ kcal/mol is one order of magnitude higher than that of $\epsilon=0.43$ kcal/mol and 0.30 kcal/mol. The mean square displacement of $\epsilon=0.50$ kcal/mol increases sharply at first, showing there are both liquid phase and 2D ice solid phase, then the inflection point appears and tends to increase slowly, indicating there are both 2D and 3D coexisting ice solid phase.

Fig.1 b Mean-square displacement of water molecules at $T=205$ K

The updated content above have been incorporated to the revised manuscript on lines 110-116.

3. Around line 124: various explanations are given, but no evidence seems to be presented. For example,

Reply: Thank you for your question. Around line 124: For $\epsilon=0.10$ kcal/mol, the formation of critical nucleus preferentially occurs inside the droplet rather than on the solid surface, which is attributed to the superhydrophobic state of the surface ($\theta > 150^\circ$). The relevant figure and data are shown in Supplementary Fig.

2 and Fig. 4. For $\epsilon=0.10$ kcal/mol, nucleation and growth occurs inside the nanodroplet and then diffuses to the outside, which belongs to homogeneous nucleation. The contact angle of nanodroplet is 163° , which is larger than 150° , so the surface property is superhydrophobic.

Supplementary Fig. 2 and Fig. 4 CA and Ice dynamic formation process

4. Line 122: They state that nucleation occurs in the droplet and not on the solid surface because the surface is superhydrophobic, but they do not show a causal relationship between the two, which reads as just an unfounded prediction.

Reply: Thank you for your question. For $\epsilon=0.10$ kcal/mol, the contact angle of nanodroplet is 163° ($\theta > 150^\circ$) in Supplementary Fig. 2, so the surface property is superhydrophobic. After quenching, nucleation occurs in the droplet and not on the solid surface as shown in Supplementary Fig. 4, which is attributed to the superhydrophobicity of the surface.

5. Line 128: “With increasing wettability, the contact angle of nanodroplet decreases and the contact radius increases, which enhances the interaction between droplet and the surface.” I think the causal relationship is the opposite.

Reply: Thank you for pointing out this issue. We have revised the sentence to “With the increase of wettability, the interaction between nanodroplet and the surface enhances, so the contact angle of nanodroplet decreases and the contact radius increases.”

The updated content above have been incorporated to the revised manuscript on lines 139-141.

6. Line 131: “Nanodroplet on surface with higher wettability is more likely to nucleate and crystallize due to the very high adsorption energy of the surface” It is just a guess and no evidence is provided.

Reply: In our previous work [1], the relationship between the nucleation rate of nanodroplets and the surface wettability was investigated. The surface energy is the nanodroplet-surface binding energy, which governs the surface wettability. The range of ϵ_{w-Pt} from 0.115 to 0.346 kcal/mol is able to cover the surface wettability from hydrophobic to hydrophilic. The hydrophilic surface attracts water molecules more easily than the hydrophobic surface owing to the strong solid-water characteristic energy. It is seen that the ice nucleation rate is slightly enhanced by increasing the energy parameter ϵ_{w-Pt} . The enhancement of ice nucleation rate is attributed to the increased ice nucleation on hydrophilic surfaces with very high adsorption energy. The heterogeneous ice nucleation can be promoted due to the nucleation on solid surfaces with very high interaction strength [2].

Fig.2 The nucleation rate as a function of surface wettability

[1] Jiang J., Li G.X., Sheng Q., Tang G.H. Microscopic mechanism of ice nucleation: The effects of surface rough structure and wettability. *Appl. Surf. Sci.* 510, 145520 (2020).

[2] M. Fitzner, G.C. Sosso, S.J. Cox, A. Michaelides. The many faces of heterogeneous ice nucleation: interplay between surface morphology and hydrophobicity. *J. Am. Chem. Soc.* 137, 13658–13669 (2015).

7. Line 134: I don't know what a “large-scale hydrogen bond” is.

Reply: Apologies for the unclear expression. We have changed “large-scale hydrogen bond” to “a large number of hydrogen bonds”. Ice nucleation occurs once a sufficient number of relatively long-lived hydrogen bonds develop spontaneously at the same location to form a fairly compact initial nucleus. The initial nucleus then slowly changes shape and size until it reaches a stage that allows rapid expansion, resulting in crystallization of the entire system [1].

Fig. 3 The hydrogen bond network structure of water [1]

It is certainly interesting to discuss the role of hydrogen bond, but we are unfortunately not able to do it using mW model. The choice of mW model here should be attributed to its high calculation efficiency, especially for ice-nucleation, a very slow process for molecular dynamics simulations. Moreover, mW model has been verified for its application in ice-nucleation.

[1] Matsumoto, M., Saito, S. & Ohmine, I. Molecular dynamics simulation of the ice nucleation and growth process leading to water freezing. *Nature* 416, 409-413 (2002).

8. Line 135: The phenomenon of molecules aligning to form crystals is not limited to water. I don't think water is particularly prone to ordering, but if you claim that, you need evidence.

Reply: Thank you for pointing out this issue. After water freezing, it usually forms stacking disordered hexagonal ice and cubic ice. We have deleted it.

9. Line 136: It is an obvious statement over 10 lines.

Reply: Thank you for your question. We have modified it to make it less than 5 lines.

The updated content above have been incorporated to the revised manuscript on lines 147-151.

10. Line 143: This is not true. The most stable phase is the phase with the lowest free energy, not the lowest energy.

Reply: Apologies for the unclear expression. Thank you for your question, which help us improve the readability of the manuscript. We have rewritten the sentence to “From the thermodynamic point of view, the most stable phase is the phase with the lowest free energy, depending on the size of the solid phase free energy and liquid phase free energy.”

The updated content above have been incorporated to the revised manuscript on lines 151-153.

11. Line 145 is correct, but it is an obvious thermodynamic fact and I am not sure what they are trying to emphasize.

Reply: Thank you for your question. We have deleted the sentence.

12. Line 147: “The nucleation and crystallization of nanodroplets mainly occur on the solid surface” Since no conditions are set and no evidence is given, it is not clear what kind of situation is envisioned in the statement.

Reply: Thank you for your question. The relevant evidence is shown in Supplementary Fig. 4 and Fig. 5. For $\epsilon=0.10$ kcal/mol, the nucleation and growth of nanodroplets occur inside the nanodroplet and then diffuses to the outside. For $\epsilon=0.20-1.0$ kcal/mol, the nucleation and crystallization of nanodroplets occur on

the solid surface. Therefore, the nucleation and crystallization of nanodroplets mainly occur on the solid surface.

Supplementary Fig. 4 and Fig. 5 Ice dynamic formation process on surfaces

REVIEWER COMMENTS

Reviewer #2 (Remarks to the Author):

After the revisions, the manuscript has undergone significant improvements. I would like to extend my appreciation to the authors for their thorough review and thoughtful incorporation of the suggested changes. Their responsiveness to the required revisions is commendable. Considering the enhancements made, I recommend the publication of this manuscript in Nature Communications.

Reviewer #3 (Remarks to the Author):

In general, there are many parts that are written as if they are facts, and it is difficult for the reader to discern them. It is necessary to clearly state whether it is a logically derived general fact, an experimental fact, or a fact discovered in this study. For example, in line 149, there is a statement that small kinetic energy causes stacking disorder, but this is a causal law that is not at all obvious to the reader, and a rationale should be provided.

There are mentions of free energy in various places, but they are just rhetoric, since no free energy calculations were actually done in this paper. Low free energy and thermodynamically stable are synonymous, and it makes no sense to substitute the latter for the former without a numerical argument. If the authors are going to discuss the free energy barriers, they should have an estimate of their height, etc. Otherwise, it is mere speculation. Such expressions are present throughout the paper. (e.g. lines 127, 151, 180, 204)

Figure 1b Two snapshots are indicated by arrows from the red line, but the meaning is unclear. Figure 1b It is difficult to understand why the graph of mean-square displacement, which is the average of many samples, bends at a particular position. The caption reads "Mean-square displacement of water molecules to the solid surface at T=205 K." It is not clear what is meant by "to the solid surface". Since the main text says "on the solid surface," I assumed that it refers to water molecules in contact with the surface at time 0, but I still do not understand the mechanism by which the diffusion velocities of all the molecules in question slow down at 0.4 ns after the start of the measurement. A more detailed explanation of this situation is needed.

Line 127 In nucleation theory, it is accepted that nucleation proceeds by crossing a free energy barrier, but it is unclear whether nucleation theory can be applied in a finite size system and in a spatially anisotropic system. What does free energy mean in such a case? If the authors are going to refer to free energy, they need to discuss based on the component values quantitatively.

Line 146 This expression is unclear and should be presented quantitatively. As far as I understand, the number of hydrogen bonds in the liquid and solid phases should not be significantly different.

Line 149 The causal relationship between the small kinetic energy and the occurrence of stacking disorders is unknown.

Line 151 Since no free energy calculations were performed in this paper, what is said here is merely a reiteration of the obvious (as described in textbooks) that when a phase transition occurs, the phase that occurs has a lower free energy. I consider it almost meaningless and misleading to mention free energy in a paper that does not perform free energy calculations or component analysis of thermodynamic quantities based on thermodynamic equations.

Line 154 Again, it is unclear whether this is a general fact or a result found in this study. It is also not clear to what degree "mainly" represents.

Line 155 Numerical data is not provided. Without a discussion of how high the energy barrier is and, in contrast, how much kinetic energy is required to cross the barrier (in which case, not all kinetic energy may be directed toward crossing the barrier), it is impossible to determine whether the statement is even correct. If such an argument has been made in previous studies, it should be cited. If it is a fact found in this study, it should be written as such and accompanied by numerical evidence.

Line 160 The definition of induction time varies from paper to paper. If the author defines induction time as the time it takes for the potential energy to begin to decrease, then it should be written as such. However, there is no mention of induction time in the text hereafter, and if only one simulation was performed for each condition, it is not possible to discuss the statistical trend of induction time, so I think this line is needless.

Line 172 "Non-rotation stacking" is not a commonly used term. What is referring to rotating in either direction? What rotates? A water molecule or the whole two-dimensional ice? If it is already defined in another paper, cite it.

Line 199 What is written in this paragraph cannot be read from Figure 2b. If "intensity" refers to the area of the peak, then adding an integrated area plot would make the claim more credible.

Line 201 It needs to be explained how one can read from Figure 2b that ice growth begins at the bottom.

Figure 2 Water that was droplet in the liquid state crystallizes as bilayer ice covering the ice surface, and after the bilayer ice can no longer grow, the droplet begins to crystallize. In this case, the ratio of the amount of water in the bilayer ice to that in the droplet should depend on the substrate surface area, and it is unlikely that what happens in the paper will always happen. I feel that the paper describes a phenomenon that occurs in one particular situation as if it can be generalized.

Fig. 2a No explanation of color coding.

Fig. 2d There is no legend for the marks.

Figure 2 shows that the droplet itself moves around by thermal fluctuations. This motion corresponds to the translation of the entire system in terms of the bulk system and should be excluded from the calculation of the diffusion coefficient. A detailed explanation of how to calculate the mean-square displacement is needed.

Response Sheet

Title: Two-dimensional bilayer ice in coexistence with three-dimensional ice without confinement

Paper number: NCOMMS-23-05786A

We are glad to see that reviewers have recommended our manuscript for publication. We would like to thank the reviewers for their constructive suggestions. In the following, we have responded point-by-point to all their comments and questions and indicated the corresponding changes made in the revised manuscript. All the changes are marked in blue in the revised manuscript.

Reviewer #2 (Remarks to the Author):

After the revisions, the manuscript has undergone significant improvements. I would like to extend my appreciation to the authors for their thorough review and thoughtful incorporation of the suggested changes. Their responsiveness to the required revisions is commendable. Considering the enhancements made, I recommend the publication of this manuscript in Nature Communications.

Reply: We are grateful for the reviewer's efforts to improve the quality of our paper.

Reviewer #3 (Remarks to the Author):

In general, there are many parts that are written as if they are facts, and it is difficult for the reader to discern them. It is necessary to clearly state whether it is a logically derived general fact, an experimental fact, or a fact discovered in this study.

Reply: We thank the reviewer's positive comments. In the following, we have addressed the reviewer's concerns and indicated our revisions in the revised manuscript and Supplementary Information (SI) accordingly.

1. For example, in line 149, there is a statement that small kinetic energy causes stacking disorder, but this is a causal law that is not at all obvious to the reader, and a rationale should be provided.

Reply: When the kinetic energy of the molecule is small, the potential energy between the molecules is sufficient to confine the molecules, so that the molecules are usually arranged in a regular crystal structure. Simulations with various water models find that ice nucleated and grown is at all sizes stacking-disordered, consisting of random sequences of cubic and hexagonal ice layers [1-5]. Our simulation results reveal that the three-dimensional ice crystal structure consists of stacking-disordered hexagonal ice and cubic ice. We have revised the sentence to “*When the kinetic energy of the molecule is small, the potential energy between the molecules is sufficient to confine the molecules, so that the molecules are usually arranged in a regular crystal structure.*”

- [1] Haji-Akbari, A. & Debenedetti, P. G. Direct calculation of ice homogeneous nucleation rate for a molecular model of water. *Proc. Natl Acad. Sci. USA* 112, 10582-10588 (2015).
- [2] Li, T., Donadio, D., Russo, G. & Galli, G. Homogeneous ice nucleation from supercooled water. *Phys. Chem. Chem. Phys.* 13, 19807-19813 (2011).
- [3] Moore, E. B. & Molinero, V. Is it cubic? Ice crystallization from deeply supercooled water. *Phys. Chem. Chem. Phys.* 13, 20008-20016 (2011).
- [4] Malkin, T. L. et al. Stacking disorder in ice I. *Phys. Chem. Chem. Phys.* 17, 60-76 (2015).
- [5] Reinhardt, A. & Doye, J. P. K. Free energy landscapes for homogeneous nucleation of ice for a monatomic water model. *J. Chem. Phys.* 136, 054501 (2012).

The updated content have been incorporated to the revised manuscript on lines 155-157.

2. There are mentions of free energy in various places, but they are just rhetoric, since no free energy calculations were actually done in this paper. Low free energy and thermodynamically stable are synonymous, and it makes no sense to substitute the latter for the former without a numerical argument. If the authors are going to discuss the free energy barriers, they should have an estimate of their height, etc. Otherwise, it is mere speculation. Such expressions are present throughout the paper. (e.g. lines 127, 151, 180, 204)

Reply: Thank you for your question. Nucleation is the process with which the formation of new phases begins and is thus a widely spread phenomenon in both nature and technology. Classical nucleation theory (CNT) was formulated 90 years ago through the contributions of Volmer and Weber, Farkas, Becker and Dö ring, and Zeldovich, on the basis of the pioneering ideas of none other than Gibbs himself. CNT was created to describe the condensation of supersaturated vapors into the liquid phase, but most of the concepts can also be applied to the crystallization of supercooled liquids and supersaturated solutions. Clusters of crystalline atoms occur within the supercooled liquid by spontaneous, infrequent fluctuations, which eventually lead the system to overcome the free energy barrier for nucleation triggering the actual crystal growth [1] as shown in Fig.1.

Fig.1 Sketch of the free energy difference ΔG_N , as a function of the crystalline nucleus size n [1]. A free energy barrier for nucleation, ΔG_N^* , must be overcome to proceed from the

(metastable) supercooled liquid state to the thermodynamically stable crystalline phase through homogeneous nucleation (purple). Heterogeneous nucleation (green) can be characterized by a lower free energy barrier, $\Delta G_{M,het}^*$, and a smaller critical nucleus size, n_{het}^* .

The existence of critical ice nucleus was first confirmed by experimental study. The experimental results are in good agreement with the predictions of the classical nucleation theory (CNT) on the critical nucleus and free energy. It is the first time to experimentally confirm the existence of the critical ice nucleus and the dependence of its size on the supercooled temperature during water freezing [2]. This study effectively clarifies the general doubts in recent decades about the effectiveness of the classical nucleation theory (CNT) in describing the critical nuclear characteristics at the atomic scale, and deepens our understanding of the microscopic mechanism for the important phase transition during water freezing.

A basic physical understanding of homogeneous and heterogeneous ice nucleation is provided by the classical nucleation theory (CNT). The CNT suggests that water molecules must form an ice nucleus of critical size (critical ice nucleus) before a water crystallization process occurs. The formation of the critical ice nucleus is associated with a free energy barrier, which needs to be overcome to trigger the further growth and crystallization of ice.

The research objectives of this paper are liquid-solid, gas-solid phase diagram prediction and de-icing mechanism analysis. The free energy barrier is calculated using the kinetic reconstruction method [3] during the ice nucleation process. This method requires two quantities: the steady-state probability distribution $P_{st}(n)$, which represents the probability of finding the system in a state with the largest ice nucleus having n ice molecules, and the mean first-passage time (MFPT), $\tau(n)$, which is defined as the average time needed for the largest nucleus to contain n molecules for the first time [4]. This method has been used to study the free energy barrier for ice nucleation [5, 6].

To obtain these quantities, more than 60 independent simulations with different initial conditions were performed by MD simulations. For each simulation, the size of the largest cluster in the system is continuously monitored at regular intervals (every 1000 time steps), and the corresponding first appearance time for each size n , $t_i(n)$, is recorded. The mean first-passage time $\tau(n)$ for each size n is simply obtained by averaging $t_i(n)$ over multiple repetitions of simulation with different initial configurations. Additionally, we obtain the steady-state probability $P_{st}(n)$: at every 1000 time steps during each repetition, we sample the size n of the largest cluster in the system. Subsequently, a histogram is constructed by counting how frequently a given size n occurs as the largest cluster. By normalizing this count with respect to the total number of sampled cluster sizes, we derive the steady-state probability distribution $P_{st}(n)$.

Supplementary Fig. 6c Free energy as a function of the cluster size n ($\beta=1/k_B T$).

Supplementary Fig. 6c depicts $\beta\Delta G$ as a function of the number of ice molecules n in the largest nucleus for this case. It can be observed that $\beta\Delta G$ increases with increasing nucleus size after the inception of a small ice nucleus. When the nucleus reaches its critical size, $n=23$, $\beta\Delta G$ approaches its maximum value. Subsequently, as the ice nucleus continues to grow, the free energy decreases and triggers further ice formation. The value $\Delta G^* = 4.67k_B T$ represents the energy difference between the maximum free energy and the minimum free energy after inception of an ice nucleus. $\Delta G^* = 4.67k_B T$ is associated with the development of the ice nucleus on the solid wall.

- [1] Gabriele C., et al. Crystal Nucleation in Liquids: Open Questions and Future Challenges in Molecular Dynamics Simulations. *Chem. Rev.* 116, 7078-7116 (2016).
- [2] Bai G.Y., Gao D., Liu Z., Zhou X., Wang J.J. Probing the critical nucleus size for ice formation with graphene oxide nanosheets. *Nature*, 576, 437-441(2019).
- [3] Wedekind J., Reguera D. Kinetic reconstruction of the free-energy landscape. *J. Phys. Chem. B* 112, 11060-11063 (2008).
- [4] Wedekind J., Strey R., Reguera D. New method to analyse simulations of activated processes. *J. Chem. Phys.* 126, 134103 (2007).
- [5] Luo S., Wang J., Li Z.G. Homogeneous ice nucleation under shear. *J. Phys. Chem. B* 124, 3701-3708 (2020).
- [6] Li C., et al. Enhancing and impeding heterogeneous ice nucleation through nanogrooves. *J. Phys. Chem. C* 122, 25992-25998 (2018).

The updated figure and the explanation above have been incorporated to the revised manuscript on line 134 and Supplementary Information on lines 68-96.

3. Figure 1b Two snapshots are indicated by arrows from the red line, but the meaning is unclear.

Reply: Thank you for your kind question. In Fig.1b, the vertical axis on the right-hand side corresponds to the mean-square displacement of $\varepsilon=0.50$ kcal/mol. The mean-square displacement of $\varepsilon=0.50$ kcal/mol is one order of magnitude higher than that of $\varepsilon=0.43$ kcal/mol and 0.30 kcal/mol. The mean square displacement of $\varepsilon=0.50$ kcal/mol increases sharply at first, showing some water molecules of the droplet diffuse rapidly along the wall to form 2D ice. In this process, the liquid water exists at the same time as the 2D ice solid. Then the inflection point appears and tends to increase slowly, indicating that the liquid water molecules nucleate and grow into 3D ice. At this time, both 2D ice solid phase and 3D ice solid phase are present. The whole freezing process has two phase transitions. The first phase transition produces 2D ice, and the second phase transition produces 3D ice. The appearance of the inflection point means that the liquid water molecules nucleate and crystallize in an instant to form 3D ice, and only the solid state exists.

The updated content above have been incorporated to the revised manuscript on lines 113-121.

4. Figure 1b It is difficult to understand why the graph of mean-square displacement, which is the average of many samples, bends at a particular position. The caption reads "Mean-square displacement of water molecules to the solid surface at $T=205$ K." It is not clear what is meant by "to the solid surface". Since the main text says "on the solid surface," I assumed that it refers to water molecules in contact with the surface at time 0, but I still do not understand the mechanism by which the diffusion velocities of all the molecules in question slow down at 0.4 ns after the start of the measurement. A more detailed explanation of this situation is needed.

Reply: Apologies for the unclear expression. We have revised the sentence to "Mean-square displacement of water molecules on the solid surface at $T=205$ K." The mean square displacement of $\epsilon=0.50$ kcal/mol increases sharply at first, showing some water molecules of the droplet diffuse rapidly along the wall to form 2D ice. In this process, the liquid water exists at the same time as the 2D ice solid. Then the inflection point appears and tends to increase slowly, indicating that the liquid water molecules nucleate and grow into 3D ice. At this time, there are both 2D ice solid phase and 3D ice solid phase. The whole freezing process has two phase transitions. The first phase transition produces 2D ice, and the second phase transition produces 3D ice. The appearance of the inflection point means that the liquid water molecules nucleate and crystallize in an instant to form 3D ice, and only the solid state exists.

The updated content have been incorporated to the revised manuscript on lines 113-121 and 128-129.

5. Line 127 In nucleation theory, it is accepted that nucleation proceeds by crossing a free energy barrier, but it is unclear whether nucleation theory can be applied in a finite size system and in a spatially anisotropic system. What does free energy mean in such

a case? If the authors are going to refer to free energy, they need to discuss based on the component values quantitatively.

Reply: Thank you for your question. According to classical nucleation theory, the free energy barrier is used to evaluate nucleation crystallization, which is the most conventional method. Some literatures have calculated and analyzed the free energy in a finite size system and in a spatially anisotropic system [1-4].

Due to the exceedingly small time and length scales involved in early stages of nucleation, atomistic computer simulations can provide unique insights into microscopic aspects of nucleation and crystallization. The dynamics over the free-energy landscape dictate time evolution of systems out of equilibrium. In many systems such as chemical reactions, phase transitions, conformational changes in biomolecules etc., this dynamics is controlled by a free-energy barrier that must be surmounted; these processes are called activated and play a crucial role in many areas of science and technology.

The evaluation of activation barriers and free-energy landscapes from simulations can be accomplished using a variety of equilibrium techniques, such as umbrella sampling or transition path sampling. However, it should be noted that these equilibrium techniques may not always suffice when studying processes that occur in an out-of-equilibrium fashion. There is no guarantee beforehand that the landscape reconstructed in an equilibrium simulation truly reflects the governing landscape of the real nonequilibrium dynamics involved in the process. The introduction of additional constraints in an equilibrium simulation or the application of bias or forces in a nonequilibrium approach can significantly alter the conditions under which the process takes place and thus distort the true energy landscape. Therefore, it is highly desirable to have a method for directly evaluating this landscape based on the actual kinetic evolution of the system without imposing any artificial external forces or constraints.

A method was presented [5] that allows for direct reconstruction of the free-energy landscape from out-of-equilibrium kinetics observed during molecular dynamics (MD) simulations, Brownian dynamics (BD), stochastic simulations, and even direct experiments. This technique only requires two key components: steady-state probability distribution and mean first-passage time. Both can be calculated through simulations as well as experimental measurements.

The free energy barrier $\Delta G(n)$ for ice nucleation is obtained adopting the method developed by Wedekind and Reguera [5]. First, the number of ice molecules is obtained based on ice molecule identification using the average bond order parameter. Here, q_3 is applied to distinguish the molecular structure of cubic and hexagonal ice from liquid water. The number of ice molecules in the largest ice cluster, n , is then used to denote the state of the system. To obtain the free energy, two quantities are required in this method. One is the steady probability $P_{st}(n)$ of the system assuming a state characterized by n . The other is the mean first passage time (MFPT), $\tau(n)$, which is the mean time the system takes to reach the state for the first time. Both $P_{st}(n)$ and $\tau(n)$ can be obtained in MD simulations and the free energy is then calculated through

$$\beta\Delta G(n) = \ln[B(n)] - \int_1^n \frac{dn'}{B(n')} + C \quad (1)$$

where $\beta = 1/(k_B T)$ with k_B is the Boltzmann constant, C is a constant, and $B(n)$ is given by

$$B(n) = \frac{1}{P_{st}(n)} \left[\int_1^n P_{st}(n') dn' - \frac{\tau(n)}{\tau(n_1)} \right] \quad (2)$$

where n_1 is the maximum value of n , which is set as 200. The dependence of MFPT on n is sigmoidal, as depicted in Fig. 6a, and can be described by^[6]

$$\tau(n) = \frac{\tau_0}{2} \left(1 + \operatorname{erf} \left((n - n^*)c \right) \right) \quad (3)$$

where τ_0 is a characteristic time, c is the local curvature around the top of the barrier, and n^* is the critical nucleus size.

To obtain these quantities, more than 60 independent simulations with different initial conditions were performed by MD simulations. For each simulation, the size of the largest cluster in the system is continuously monitored at regular intervals (every 1000 time steps), and the corresponding first appearance time for each size n , $t_i(n)$, is recorded. The mean first-passage time $\tau(n)$ for each size n is simply obtained by averaging $t_i(n)$ over multiple repetitions of simulation with different initial configurations. Additionally, we obtain the steady-state probability $P_{st}(n)$: at every 1000 time steps during each repetition, we sample the size n of the largest cluster in the system. Subsequently, a histogram is constructed by counting how frequently a given size n occurs as the largest cluster. By normalizing this count with respect to the total number of sampled cluster sizes, we derive the steady-state probability distribution $P_{st}(n)$.

Supplementary Fig. 6 The calculation of ice nucleation free energy barrier at $T=205\text{K}$ ($\epsilon=0.4\text{kcal/mol}$). **a** Mean first-passage time (MFPT) $\tau(n)$ as a function of the cluster size n obtained from the MD simulations. The red line represents a fit of Eq.3. **b** Growth curves of the largest cluster from the MD simulations. **c** Kinetic reconstruction of the free energy of cluster formation obtained from the MD simulations ($\beta=1/k_B T$).

Fig.6a shows that the MFPT curve has a sigmoidal shape that reaches a well-defined plateau at larger values of n , indicating that overcoming critical size is rate-limiting step in cluster formation and subsequent growth of cluster takes negligible time compared to activation time. Fig.6b illustrates the growth curves

of the largest ice cluster from the MD simulations. The free-energy barrier ΔG are calculated through Eqs.1 and 2. Fig. 6c depicts $\beta\Delta G$ as a function of the number of ice molecules n in the largest nucleus for this case ($\varepsilon=0.4\text{kcal/mol}$). It can be observed that $\beta\Delta G$ increases with increasing nucleus size after the inception of a small ice nucleus. It approaches its maximum when the nucleus reaches the critical size, $n=23$. The free energy then decreases with increasing ice nucleus and triggers further ice formation. The value $\Delta G^* = 4.67k_B T$ represents the energy difference between the maximum free energy and the minimum free energy after inception of an ice nucleus. $\Delta G^* = 4.67k_B T$ is associated with the development of the ice nucleus on the solid wall.

[1] Qiu Y.Q., et al. Ice nucleation efficiency of hydroxylated organic surfaces is controlled by their structural fluctuations and mismatch to ice. *J. Am. Chem. Soc.* 139, 3052-3064 (2017).

[2] Bi, Y.F., Cao, B.X. & Li, T.S. Enhanced heterogeneous ice nucleation by special surface geometry. *Nat. Commun.* 8, 15372 (2017).

[3] Luo S., Wang J., Li Z.G. Homogeneous ice nucleation under shear. *J. Phys. Chem. B* 124, 3701-3708 (2020).

[4] Li C., et al. Enhancing and impeding heterogeneous ice nucleation through nanogrooves. *J. Phys. Chem. C* 122, 25992-25998 (2018).

[5] Wedekind J., Reguera D. Kinetic reconstruction of the free-energy landscape. *J. Phys. Chem. B* 112, 11060-11063 (2008).

[6] Wedekind J., Strey R., Reguera D. New method to analyse simulations of activated processes. *J. Chem. Phys.* 126, 134103 (2007).

The updated figure and the explanation above have been incorporated to the revised manuscript on line 134 and Supplementary Information on lines 68-96.

6. Line 146 This expression is unclear and should be presented quantitatively. As far as I understand, the number of hydrogen bonds in the liquid and solid phases should not be significantly different.

Reply: Apologies for the unclear expression. We have changed “a large number of hydrogen bonds” to “a relatively long-lived hydrogen-bonded network”. Ice nucleation occurs once a sufficient number of relatively long-lived hydrogen

bonds develop spontaneously at the same location to form a fairly compact initial nucleus. The initial nucleus then slowly changes shape and size until it reaches a stage that allows rapid expansion, resulting in crystallization of the entire system [1]. The ice crystal has a long-range ordered structure, while water does not have a long-range ordered structure. Only some water molecules maintain short-range order with neighboring water molecules. This orderly “water cluster” is constantly generating, destroying and moving, and its degree is related to temperature. The fluidity of water molecules in liquid water is too strong to form a large number of stable hydrogen bonds like ice. We note that the enthalpy of fusion of ice at atmospheric pressure is 6.01 kJ/mol, whilst the enthalpy of vaporisation at the triple point is 51.059 kJ/mol; in the latter process, hydrogen bonds must clearly be broken, but the low enthalpy of fusion suggests that the majority of hydrogen bonds survive in the liquid phase, although perhaps on a transient timescale [2].

Fig. 3 Number of water molecules having long-lasting hydrogen bonds and the size of the largest cluster in inherent structures [1]

It is certainly interesting to discuss the role of hydrogen bond, but we are unfortunately not able to do it using mW model. The choice of mW model here should be attributed to its high calculation efficiency, especially for ice-nucleation, a very slow process for molecular dynamics simulations.

[1] Matsumoto, M., Saito, S. & Ohmine, I. Molecular dynamics simulation of the ice nucleation and growth process leading to water freezing. *Nature* 416, 409-413 (2002).

[2] Debenedetti, P.G. ‘Supercooled and glassy water’. *J Phys: Condens Mat* 15, R1669-R1726 (2003).

The updated content have been incorporated to the revised manuscript on lines 152-153.

7. Line 149 The causal relationship between the small kinetic energy and the occurrence of stacking disorders is unknown.

Reply: Apologies for the unclear expression. We have rewritten the sentence to “When the kinetic energy of the molecule is small, the potential energy between the molecules is sufficient to confine the molecules, so that the molecules are usually arranged in a regular crystal structure.”

The updated content have been incorporated to the revised manuscript on lines 155-157.

8. Line 151 Since no free energy calculations were performed in this paper, what is said here is merely a reiteration of the obvious (as described in textbooks) that when a phase transition occurs, the phase that occurs has a lower free energy. I consider it almost meaningless and misleading to mention free energy in a paper that does not perform free energy calculations or component analysis of thermodynamic quantities based on thermodynamic equations.

Reply: Thank you for your kind question. We have deleted the sentence.

9. Line 154 Again, it is unclear whether this is a general fact or a result found in this study. It is also not clear to what degree "mainly" represents.

Reply: Thank you for your question. This is a result found in our simulation study. The relevant evidence is shown in Supplementary Fig. 4 and Fig. 5. For $\varepsilon=0.10$ kcal/mol, the nucleation and growth of nanodroplets occur inside the nanodroplet and then diffuses to the outside. For $\varepsilon=0.20-1.0$ kcal/mol, the nucleation and crystallization of nanodroplets occur on the solid surface. Therefore, the nucleation and crystallization of nanodroplets mainly occur on

the solid surface. "mainly" represents that the nucleation and crystallization of nanodroplets occur on the solid surface of $\varepsilon=0.20-1.0$ kcal/mol.

Supplementary Fig. 4 and Fig. 5 Ice dynamic formation process on surfaces

The updated content have been incorporated to the revised manuscript on line 158.

10. Line 155 Numerical data is not provided. Without a discussion of how high the energy barrier is and, in contrast, how much kinetic energy is required to cross the barrier (in which case, not all kinetic energy may be directed toward crossing the barrier), it is impossible to determine whether the statement is even correct. If such an argument has been made in previous studies, it should be cited. If it is a fact found in this study, it should be written as such and accompanied by numerical evidence.

Reply: We have revised the sentence to “the latent heat released by local crystallization can provide enough energy for growth and crystallization [53].” The relevant reference [53] have been cited to reinforce the conclusions.

The updated content have been incorporated to the revised manuscript on lines 159-160.

11. Line 160 The definition of induction time varies from paper to paper. If the author defines induction time as the time it takes for the potential energy to begin to decrease, then it should be written as such. However, there is no mention of induction time in the text hereafter, and if only one simulation was performed for each condition, it is not possible to discuss the statistical trend of induction time, so I think this line is needless.

Reply: Thank you for your kind question. We have deleted the sentence.

12. Line 172 “Non-rotation stacking” is not a commonly used term. What is referring to rotating in either direction? What rotates? A water molecule or the whole two-dimensional ice? If it is already defined in another paper, cite it.

Reply: Apologies for the unclear expression. Thank you for your question, which help us improve the readability of the manuscript. “Non-rotation stacking” means that the positions of water molecules in the upper and lower layers of 2D ice are completely coincident. We have revised the sentence to “The 2D ice is formed by two layers of interlocking 5-, 6-, 7- membered rings water molecules.”

The updated content have been incorporated to the revised manuscript on lines 175-176.

13. Line 199 What is written in this paragraph cannot be read from Figure 2b. If "intensity" refers to the area of the peak, then adding an integrated area plot would make the claim more credible.

Reply: Apologies for the unclear expression. In Fig.2b, “intensity” represents the size of density. “Height” is the size of z axis. The intensity of the first peak is slightly larger than that of the second peak, indicating that the growth of 2D ice starts from the bottom layer (near the solid wall).

The updated content have been incorporated to the revised manuscript on lines 202-205 and 214.

14. Line 201 It needs to be explained how one can read from Figure 2b that ice growth begins at the bottom.

Reply: Apologies for the unclear expression. We have revised the sentence to “The intensity of the first peak is slightly larger than that of the second peak as shown in Fig. 2b, showing that the growth of 2D ice starts from the bottom layer (near the solid wall), which is consistent with the results in Reference [40]”.

The updated content have been incorporated to the revised manuscript on lines 202-205.

15. Figure 2 Water that was droplet in the liquid state crystallizes as bilayer ice covering the ice surface, and after the bilayer ice can no longer grow, the droplet begins to crystallize. In this case, the ratio of the amount of water in the bilayer ice to that in the droplet should depend on the substrate surface area, and it is unlikely that what happens in the paper will always happen. I feel that the paper describes a phenomenon that occurs in one particular situation as if it can be generalized.

Reply: Thank you for your question. We have studied many working conditions, found this new phenomenon occurring under all the conditions. At present, the knowledge on 2D-3D coexisting ice formation and de-icing characteristics at molecular level remains blank. The influences of the surface wettability, temperature, pressure and phase transition mode on the formation of 2D-3D coexisting ice without confinement has not been studied. Our findings fill the gap about the coexistence of 2D bilayer ice and 3D ice without nanoscale confinement. We explore the possibility of controlling the formation of low- and high-dimensional ices by tuning the interactions of a water droplet on a surface. Our work will hopefully motivate future experimental studies of the formation of low- and high-dimensional ice on realistic surfaces.

16. Fig. 2a No explanation of color coding.

Reply: Thank you for your kind question. We have added the sentence “Dark yellow, light blue, light gray (3.08ns) and light red balls represent hexagonal ice, cubic ice, 2D ice and solid substrate, respectively”.

The updated content have been incorporated to the revised manuscript on lines 212-213, 261-263 and 446-447.

17. Fig. 2d There is no legend for the marks.

Reply: Thank you for your kind question. We have added the sentence “Phase diagram consists of liquid water, 2D ice in coexistence with liquid water, 3D ice and 2D-3D coexisting ice.”

The updated content have been incorporated to the revised manuscript on lines 216-217, 264 and 449-451.

18. Figure 2 shows that the droplet itself moves around by thermal fluctuations. This motion corresponds to the translation of the entire system in terms of the bulk system and should be excluded from the calculation of the diffusion coefficient. A detailed explanation of how to calculate the mean-square displacement is needed.

Reply: Thank you for your question. Our research ignores the influence of thermal fluctuation on the mean square displacement, and we refer to the relevant literature [1, 2] for calculation and analysis. The mean square displacement (MSD) at time t is calculated by Eq. (5):

$$\text{MSD}(t) = \frac{1}{n} \sum_{i=0}^n (r_i^t - r_i^0)^2 \quad (5)$$

where r_i^t and r_i^0 are the positions of atom i at time t and 0, respectively.

[1] Zhu, C.Q. et al. Direct observation of 2-dimensional ices on different surfaces near room temperature without confinement. *Proc. Natl. Acad. Sci. USA* 116, 16723-16728 (2019).

[2] Luo S., Wang J., Li Z.G. Homogeneous ice nucleation under shear. *J. Phys. Chem. B* 124, 3701-3708 (2020).

The updated content have been incorporated to the revised manuscript on lines 511-514.

REVIEWERS' COMMENTS

Reviewer #2 (Remarks to the Author):

The manuscript has seen considerable enhancements as a result of the revisions made by the authors in response to the reviewer's feedback. The authors have diligently reviewed every comment and suggestion put forth by the reviewer. These modifications have greatly enhanced the original work. With these improvements in mind, I enthusiastically endorse the publication of this manuscript in Nature Communications.